# Domain fusion TLR2-4 enhances the autophagy-dependent clearance of *Staphylococcus aureus* in the genetic engineering goat

Mengyao Wang[1,2,3†], Yu Qi[1,2,3†], Yutao Cao[1,2,3], Xiaosheng Zhang[4], Yongsheng Wang[5], Qingyou Liu[6], Jinlong Zhang[4], Guangbin Zhou[7], Yue Ai[1,2,3], Shao Wei[1,2,3], Linli Wang[1,2,3], Guoshi Liu[1,2,3], Zhengxing Lian[1,2,3], Hongbing Han[1,2,3]*

[1]Beijing Key Laboratory of Animal Genetic Improvement, College of Animal Science and Technology, China Agricultural University, Beijing, China; [2]National Engineering Laboratory for Animal Breeding, College of Animal Science and Technology, China Agricultural University, Beijing, China; [3]Key Laboratory of Animal Genetics, Breeding and Reproduction of the Ministry of Agriculture and Rural Affairs, College of Animal Science and Technology, China Agricultural University, Beijing, China; [4]Tianjin Academy of Agricultural Sciences, Tianjin, China; [5]Key Laboratory of Animal Biotechnology, Ministry of Agriculture, Northwest Agriculture and Forest University, Shaanxi, China; [6]State Key Laboratory for Conservation and Utilization of Subtropical Agro-bioresources, Guangxi University, Nanning, China; [7]Farm Animal Genetic Resources Exploration and Innovation Key Laboratory of Sichuan Province, College of Animal Science and Technology, Sichuan Agricultural University, Chengdu, China

**\*For correspondence:**
hanhongbing@cau.edu.cn

†These authors contributed equally to this work

**Abstract** *Staphylococcus aureus* infections pose a potential threat to livestock production and public health. A novel strategy is needed to control *S. aureus* infections due to its adaptive evolution to antibiotics. Autophagy plays a key role in degrading bacteria for innate immune cells. In order to promote *S. aureus* clearance via Toll-like receptor (TLR)-induced autophagy pathway, the domain fusion TLR2-4 with the extracellular domain of TLR2, specific recognizing *S. aureus*, and transmembrane and intracellular domains of TLR4 is assembled, then the goat expressing *TLR2-4* is generated. TLR2-4 substantially augments the removal of *S. aureus* within macrophages by elevating autophagy level. Phosphorylated JNK and ERK1/2 promote LC3-puncta in TLR2-4 macrophages during *S. aureus*-induced autophagy via MyD88 mediated the TAK1 signaling cascade. Meantime, the TRIF-dependent TBK1-TFEB-OPTN signaling is involved in TLR2-4-triggered autophagy after *S. aureus* challenge. Moreover, the transcript of *ATG5* and *ATG12* is significantly increased via cAMP-PKA-NF-κB signaling, which facilitates *S. aureus*-induced autophagy in TLR2-4 macrophages. Overall, the novel receptor TLR2-4 enhances the autophagy-dependent clearance of *S. aureus* in macrophages via TAK1/TBK1-JNK/ERK, TBK1-TFEB-OPTN, and cAMP-PKA-NF-κB-ATGs signaling pathways, which provide an alternative approach for resistant against *S. aureus* infection.

## Editor's evaluation

In this manuscript, the authors generate a goat expressing a domain fusion receptor TLR2-4 that allows macrophages from the genetically modified goat to eliminate *Staphylococcus aureus*. This study is of interest to animal geneticists studying molecular breeding for infection resistance and

individuals interested in bacterial infections. These studies also serve as a good animal model for disease resistance breeding.

## Introduction

*Staphylococcus aureus (S, aureus)* is regarded as an important zoonotic pathogen that causes disease in both human and livestock (*Fluit, 2012*; *Tong et al., 2015*). *S. aureus* usually gives rise to skin and soft tissue infections, pneumonia, device-related infections, severe sepsis, etc. (*Lee et al., 2018*; *Lowy, 1998*). In addition, *S. aureus*-induced mastitis is a major disease that poses a potential harm to animal welfare in global dairy industry. Moreover, the transmission of *S. aureus* from livestock to dairy products can lead to human food poisoning (*Hennekinne et al., 2012*; *Mechesso et al., 2021*). However, *S. aureus* has developed antibiotic resistance mechanisms for most kinds of antibiotics due to the widespread use of antibiotic (*Pantosti et al., 2007*). It is urgent to develop new approaches to overcome the *S. aureus* infection.

The innate immune system is the first line for defense against invading pathogens. Toll-like receptor (TLR), one of pattern recognition receptor, can sense pathogen-associated molecular patterns (PAMPs) from various pathogens to trigger immune responses (*Kawai and Akira, 2007*). It has been proved that TLR2 plays an essential role in mediating immune response to Gram-positive bacteria, including *S. aureus*, by forming a heterodimerize with TLR6 or TLR1 (*Akira and Takeda, 2004*; *Takeuchi et al., 2001*; *Takeuchi et al., 2002*). The extracellular domain of TLR2 contains 18–20 leucine rich repeat (LRR) and LRR-like motifs, which is responsible for recognizing *S. aureus* (*Kirschning and Schumann, 2002*). The interaction between the intracellular domain of TLR2 and downstream signaling adaptor MyD88 induces phosphorylation of mitogen-activated protein kinases (MAPK) and activation of NF-κB, resulting in production of proinflammatory cytokines, chemokines, and costimulatory molecule involved in acquired immune responses to eliminate *S. aureus* (*Bannerman et al., 2004*; *Brenaut et al., 2014*; *Bulgari et al., 2017*).

TLR signal also enhances the killing of invading pathogens via activating autophagy pathways (*Sanjuan et al., 2007*). Autophagy is an evolutionarily conserved intracellular process by which cytosolic material is engulfed in a double-membrane vesicle termed autophagosome, followed by fusion with lysosomes to form autophagosomes for degradation and recycling (*Levine and Kroemer, 2019*). In addition to the conventional homeostatic function, autophagy also plays critical roles in immunity by regulating cytokine production, antigen presentation, removal of dangerous cargo (*Levine and Kroemer, 2008*; *Levine and Kroemer, 2019*). A publication has shown that autophagy plays a protective role against intracellular *S. aureus* at low multiplicity of infection (MOI) in murine fibroblasts, where *S. aureus* was rapidly ubiquitinated and restricted (*Neumann et al., 2016*). Knockdown of autophagy receptor sequestosome 1 (SQSTM1/p62) significantly increased the cytosolic *S. aureus* burden within host cell (*Gibson et al., 2020*). Simultaneously, autophagic pathway induced by TLR4 facilitates clearance of internalized bacteria residing in the cytosol, including *Mycobacterium tuberculosis* and *Salmonella typhimurium* (*Liu et al., 2019*; *Pahari et al., 2020*; *Xu et al., 2007*). TLR4 recognizing lipopolysaccharide from Gram-negative bacteria induces the cytosolic Toll/interleukin-1 receptor (TIR) domains dimerization, then activates TIR-containing adaptor protein to initiate autophagy (*Liu et al., 2019*; *Xu et al., 2007*). The coordination of TLR2 and TLR4 signal triggers innate immune responses in the mammary epithelial cells to contribute to counteracting *S. aureus* infections (*Goldammer T, Zerbe H, Molenaar A, Schuberth HJ, Brunner RM, Kata SR, and Seyfert HM, 2004*). Further, TLR2/TLR4-dependent autophagy also enhances the phagocytosis and destruction of *S. aureus* in mouse macrophages (*Shu et al., 2020*).

We hypothesized that *S. aureus* could be extremely eliminated in macrophages by regulating autophagy pathway with combined TLR2 recognizing *S. aureus* and TLR4 activating autophagy. Since activation of TLR2 signal needs to form a heterodimer with TLR1 or TLR6, only the overexpression of TLR2 can unlikely enhance *S. aureus*-induced autophagy. However, TLR4, as a oneself homodimer, can induce autophagy pathways by the interaction of its intracellular domains and other adaptor molecule. So, we establish a recombind gene *TLR2-4* by fusing extracellular domain of TLR2 (specific recognizing *S. aureus*) with transmembrane and intracellular domains of TLR4 (initiating autophagy pathways). The goat expressing TLR2-4 was generated by clustered regularly interspaced short palindromic repeats/CRISPR-associated system 9 (CRISPR/Cas9) mediated knock-in. The expression of

LC3-II in macrophages from TLR2-4 goat was obviously increased, and bacterial burden was significantly reduced after *S. aureus* challenge. Moreover, the activation of autophagy pathway in TLR2-4 macrophages relied on JNK/ERK, cAMP-PKA, and TBK1-TFEB-OPTN signaling pathways. Taken together, we identified that domain fusion receptor TLR2-4 could recognize *S. aureus* and enhance autophagy-mediated clearance of *S. aureus* in macrophages. The present study is, to our knowledge, the first research describing the fusion gene *TLR2-4* in dairy ruminants.

## Results
### Establishment of the fibroblast cell line expressing TLR2-4

The fusion gene *TLR2-4* was assembled with the coding sequence (CDS) of goat *TLR2* extracellular domain including a signal peptide and ligand recognition domain and *TLR4* transmembrane and intracellular domain by overlap PCR (*Figure 1A*). To easily detect the expression of *TLR2-4* in fibroblast cell, MyC-tag CDS was introduced between signal peptide and ligand recognition CDS of TLR2 (*Figure 1A*). Eight CRISPR/Cas9 small guide RNA (sgRNA) targets were screened within the first intron of SET domain-containing 5 (*SETD5*) for achieving site-specific integration of *TLR2-4* by T7 endonuclease 1 (T7E1) assay (*Figure 1—figure supplement 1A*), and sgRNA2 was selected according to the genome editing efficiency (*Figure 1B*). The *TLR2-4* expression vector (containing *tdtomato* and ~1 kb homologous arm) and the sgRNA2-Cas9 expression vector (PX458, containing *GFP*) were co-transfected into goat fibroblasts, which might complete site-specific insertion of *TLR2-4* by CRISPR/Cas9-induced homology-mediated end joining (HMEJ) (*Figure 1C*). After three days, cells expressing both *GFP* and *tdtomato* were selected by flow cytometry sorting, then the monoclonal cells were obtained after cultured for 14 days (*Figure 1—figure supplement 1B*). The fibroblasts of site-specific integration of *TLR2-4* were identified by PCR, and the DNA fragment of 1420 bp was further purified to sequence by Sanger sequencing (*Figure 1—figure supplement 1C*). The efficiency of targeted knock-in of *TLR2-4* was 19.05% (*Figure 1—figure supplement 1D*). Further, the expression of TLR2-4 was detected by Western blot (*Figure 1D*, *Figure 1—source data 1*), immunofluorescence (*Figure 1E*), and flow cytometry using anti-MyC tag antibody (*Figure 1—figure supplement 1E*), indicating that the expression of TLR2-4 was located on cytomembrane and cytoplasm. To confirm whether TLR2-4 could execute its functions, the TLR2-4 fibroblast cells that endogenous *TLR2* was knockdown with small interfering RNA (siRNA, knockdown efficiency of *TLR2* is about 80%, *Figure 1—figure supplement 1F*) were stimulated by TLR2 agonist Pam2CSK4 (P2C), Pam3CSK4 (P3C), and heat-killed *S. aureus* (HK-SA), respectively. Compared to negative control siRNA, the expression of IL-6, IL-8, and IL-1β was decreased in TLR2-4 and wild-type (WT) fibroblast cells with siRNA of *TLR2* after HK-SA treatment, receptively (*Figure 1—figure supplement 1G,H*). The mRNA level of IL-6 in TLR2-4 fibroblast cells with siTLR2 was dramatically increased with P2C and P3C treatment (*Figure 1F*). Moreover, the mRNA levels of IL-6 and IL-8 in TLR2-4 fibroblast cells with siRNA TLR2 were dramatically increased with HK-SA treatment (*Figure 1G*). These results indicated that the fusion gene *TLR2-4* can recognize ligands of TLR2 to initiate immune response.

Meantime, TLR2-4 and WT fibroblast cells that endogenous *TLR2* was knockdown by siRNA were challenged by HK-SA, and then transcriptomic analysis showed 557 differentially expressed genes (DEGs), compared to WT fibroblast cells, 409 of which were up-regulated, whereas 148 genes were down-regulated (*Figure 1—figure supplement 2A*). Gene Ontology (GO) enrichment analysis revealed that the significant terms of DEGs were mainly involved in cytokine activity, NF-κB signaling, and MAPK activity (*Figure 1H*). In addition, the Kyoto Encyclopedia of Genes and Genomes (KEGG) pathway analysis revealed that the DEGs were mainly enriched in forkhead box O (FoxO) signaling pathway, MAPK signaling pathway, and JUN kinase pathway (*Figure 1—figure supplement 2B*).

To confirm the data of RNA-seq, we investigated the TLR2-4-related downstream signaling pathways involved in inflammatory cytokine, including the MAPK signaling pathways and NF-κB signaling pathway. The phosphorylation of ERK1/2 and JNK in TLR2-4 fibroblasts with HK-SA treatment was substantially increased (*Figure 1I*, *Figure 1—source data 2*). In addition, the level of NF-κB subunit p65 translocation was almost suppressed completely in TLR2-4 fibroblasts independent of *S. aureus* treatment (*Figure 1I*). These data suggested that TLR2-4 in goat fibroblast cells can induce the activation of JNK and ERK signaling in case of recognizing *S. aureus*, while inhibiting the NF-κB signaling.

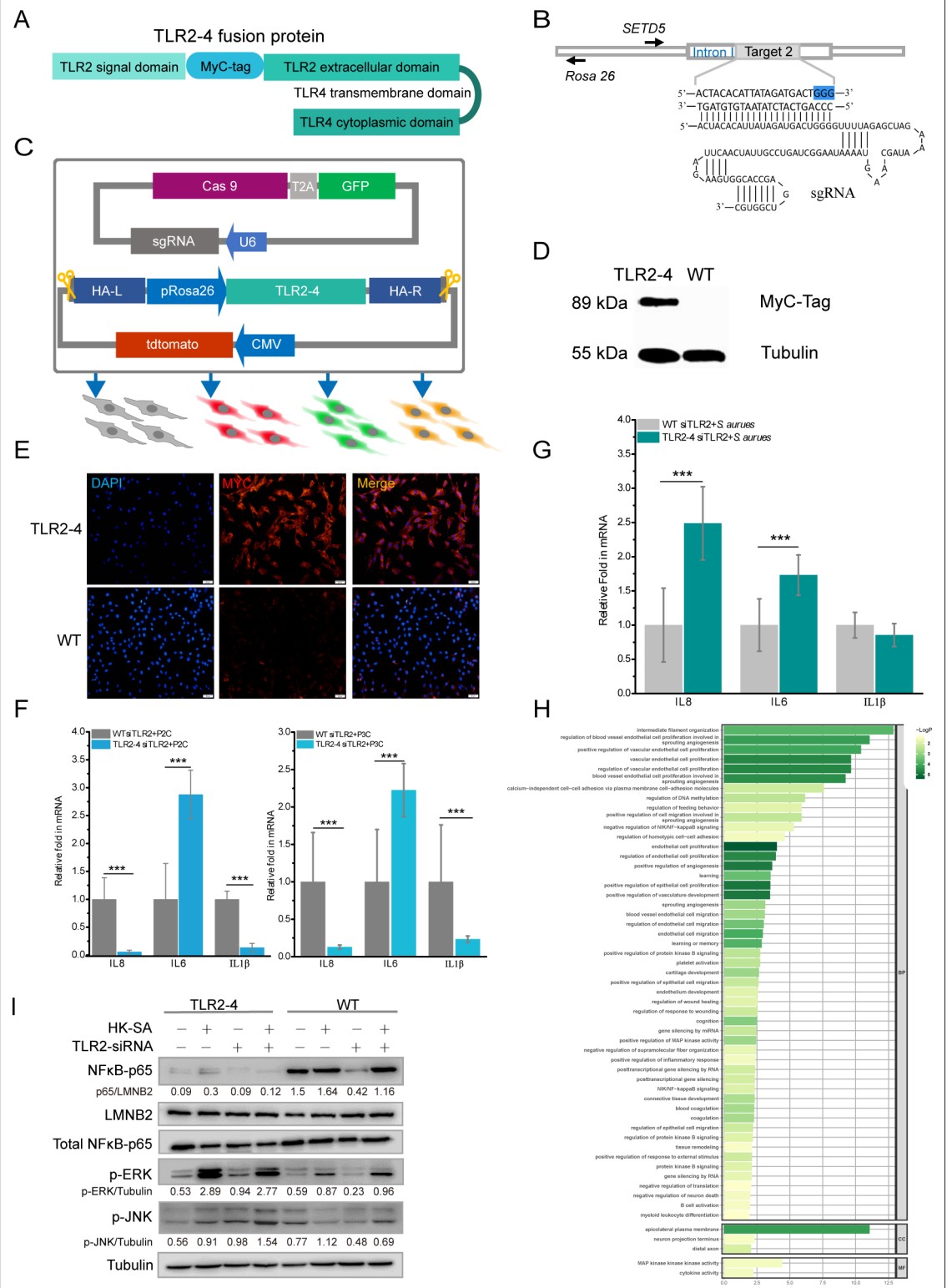

**Figure 1.** Construction of fibroblast cell line expressing Toll-like receptor 2-4 (TLR2-4). (**A**) Structure of fusion protein TLR2-4. Total RNA was extracted from goat peripheral blood and was reverse- transcribed into cDNA. TLR2 extracellular domain and TLR4 transmembrane and intracellular domain were amplified from cDNA. (**B**) Schematic representation of the small guide RNA2 (sgRNA2) and its target genome DNA. The sgRNA target 2 was located on the first intron of *SETD5* and the PAM site is colored blue. (**C**) The expression vector of sgRNA2-CRISPR/Cas9 was constructed by joining the crRNA-oligo into PX458 vector containing GFP. TLR2-4 was inserted into pRosa26-promoter vector, and the expression construction of CMV-tdTomato was cloned into reconstructed pRosa26-TLR2-4 vector. Both PX458-sgRNA and pRosa26-TLR2-4-tdTomato were co-transfected into goat fibroblasts.

*Figure 1 continued on next page*

*Figure 1 continued*

Transfected cells expressing both green and red fluorescence were selected by flow cytometry. (**D**) The protein level of TLR2-4 in fibroblasts was detected by Western blotting using MyC-Tag mouse monoclonal antibody. Tubulin was used to normalize the data of each sample. (**E**) The expression of TLR2-4 was confirmed by immunofluorescence in fibroblasts. The selected positive and wild-type (WT) fibroblasts were incubated with MyC-Tag primary antibody and conjugated with Alexa Fluor 594-labeled goat anti-mouse secondary antibody, then were observed by confocal microscopy. Scale bar: 50 µm. (**F and G**) The mRNA expression of cytokines was detected by real-time reverse transcription PCR (qRT-PCR) in TLR2 ligands or killed *S. aureus*-stimulated TLR2-4 and WT fibroblasts. Cells were pretreated with small interfering RNA (siRNA) of *TLR2* for 24 hr, and then treated with 500 ng/ml P2C, P3C (**F**) or heat-killed *S. aureus* (HK-SA) (multiplicity of infection [MOI] = 10) (**G**) (n=6 biologically independent samples). Cells were harvested at 10 hr after infection, and the mRNA levels of IL8, IL6, and IL1β were measured by qRT-PCR. (**H**) Gene Ontology (GO) enrichment terms of differentially expressed genes (DEGs) between both endogenous TLR2 silenced TLR2-4 and WT fibroblasts after stimulated with HK-SA (MOI = 10) for 10 hr according to RNA-seq data. (**I**) The cells were challenged with HK-SA (MOI = 10) for 10 hr, and the phosphorylation of ERK1/2 and JNK, and the nuclear translocation of NF-$\kappa$B were detected by Western blot. The results are presented as the means SD (standard deviation). Student's t tests were used for comparisons between two groups. ***p<0.001. Exact p-values are provided in *Supplementary file 1*.

The online version of this article includes the following source data and figure supplement(s) for figure 1:

**Source data 1.** The original blots of Toll-like receptor 2-4 (TLR2-4) (TLR2-4 blot was repeated twice) and wild-type (WT) fibroblasts.

**Source data 2.** The original blots.

**Figure supplement 1.** Establishment and identification of the Toll-like receptor 2-4 (TLR2-4) fibroblast cell line.

**Figure supplement 1—source data 1.** The original gels of *Figure 1—figure supplement 1A*.

**Figure supplement 1—source data 2.** The original gels of *Figure 1—figure supplement 1C*.

**Figure supplement 1—source data 3.** The original gels of *Figure 1—figure supplement 1C*.

**Figure supplement 2.** Heatmap and Kyoto Encyclopedia of Genes and Genomes (KEGG) pathway enrichment analysis of the differentially expressed genes (DEGs) between Toll-like receptor 2-4 (TLR2-4) and wild-type (WT) fibroblasts after *S. aureus* infection (n=3 biologically independent samples).

## Generation of clone goat expressing TLR2-4

One-hundred and twenty-six of reconstructed embryos prepared by TLR2-4 fibroblasts nuclear transfer were transplanted into twelve recipient goats with estrus synchronization, two of which became pregnant, and then one recipient was delivered ultimately (*Supplementary file 2A*). One clone goat was obtained (*Figure 2A*). The site-specific integration of *TLR2-4* in peripheral blood macrophages from clone goat was determined by PCR (*Figure 2B*). The1482 and 2700 bp fragments were amplified using primers S-M and primers R-T, respectively (*Figure 2C*, *Figure 2—source data 1*). Sanger sequencing showed that *TLR2-4* was inserted into the first intron of *SETD5*, and the 2700 bp fragment contains a 1034 bp repetitive homologous arm sequence, indicating the integration of 3' of *TLR2-4* was occured by the HMEJ. The integration of TLR2-4 was further detected by primers U-L and U-L1 (*Figure 2D*). A 1188 bp exogenous fragment that only existed in TLR2-4 macrophages was amplified using primers U-L1. A 3795 bp exogenous fragment and a 312 bp endogenous fragment were amplified by primers U-L, receptively (*Figure 2E*, *Figure 2—source data 2*), indicating *TLR2-4* was a heterozygous allele in clone goat. Sanger sequencing further showed that the expression construction of *TLR2-4* was intact (*Figure 2—figure supplement 1A*). Next, the mRNA expression of *TLR2-4* in macrophages were confirmed by RT-PCR (*Figure 2F*, *Figure 2—source data 3*). Meanwhile, the expression of TLR2-4 was detected by Western blot using MyC-tag antibody (*Figure 2G*, *Figure 2—source data 4*), flow cytometry (*Figure 2H*), and immunofluorescence assays (*Figure 2I*), which indicated the expression of TLR2-4 was located on cytomembrane and cytoplasm in clone goat macrophages. Additionally, the higher production of *IL-6* and *IL-8* was induced in TLR2-4 macrophages, compared to WT macrophages stimulated with *S. aureus* (*Figure 2—figure supplement 1B*), indicating that TLR2-4 can complete the functions during immune responses in macrophages.

## TLR2-4 promotes autophagy-dependent clearance of *S. aureus* in macrophages

Macrophages play an essential role in phagocytizing pathogenic bacteria. We evaluated the phagocytic capacity of TLR2-4 macrophages for *S. aureus* by flow cytometry. The infection rates of *S. aureus* in TLR2-4 macrophages were distinctly higher than WT macrophages at different time after live *S. aureus* challenge, regardless the MOI 1 or 10 (*Figure 2—figure supplement 1C, D*). Meanwhile, infection index of *S. aureus* was also significantly increased in TLR2-4 macrophages (*Figure 3A and B* and *Figure 2—figure supplement 1E, F*). In other words, the number of phagocytic *S. aureus* of each

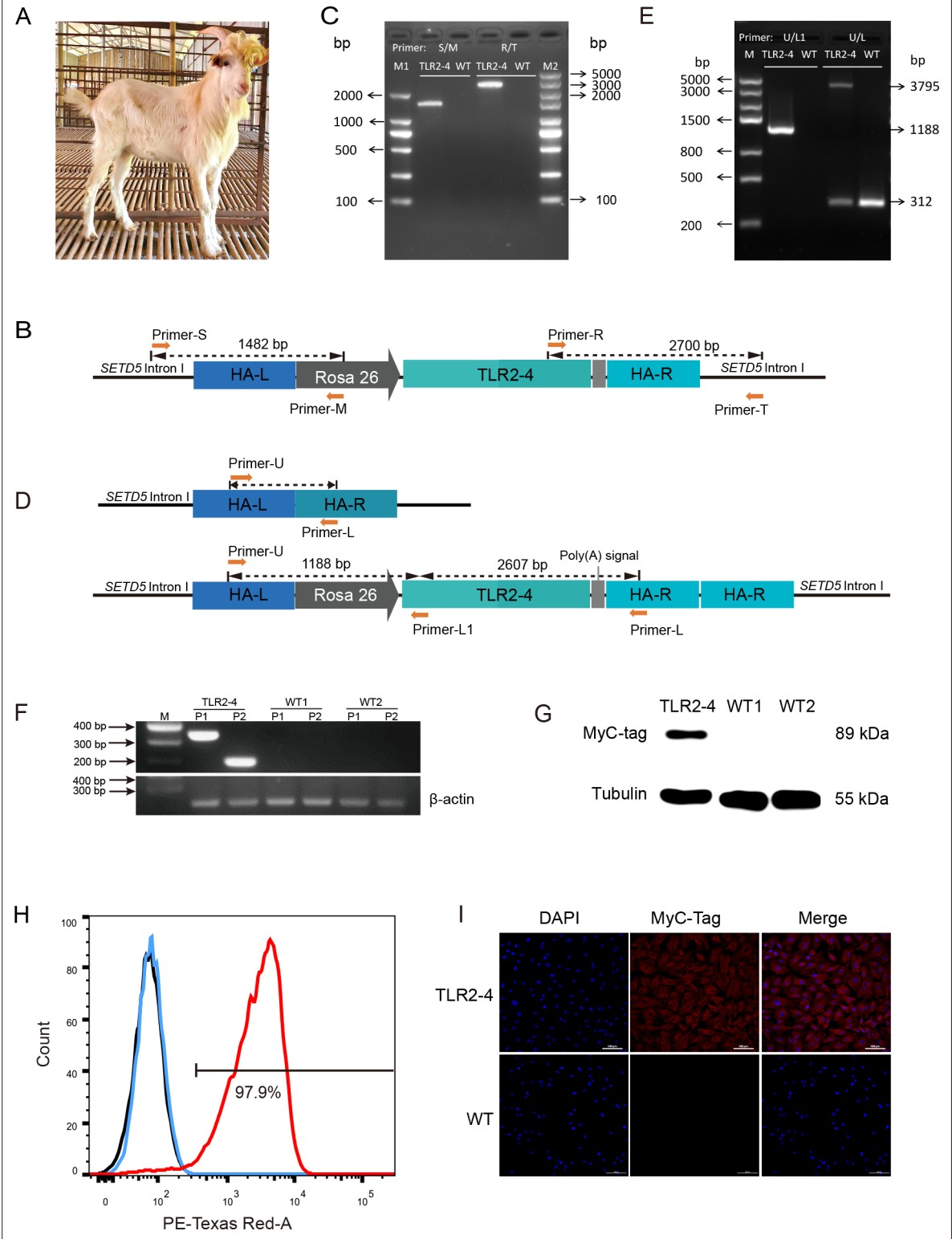

**Figure 2.** Generation of clone goat expressing Toll-like receptor 2-4 (TLR2-4). (**A**) The clone goat expressing TLR2-4 was prepared by somatic cell nuclear transfer. (**B**) Schematic representation of the site-specific integration of primers S-M and T-R. (**C**) Genomic DNA of macrophages from transgenic and wild-type (WT) goat was extracted. The PCR products were amplified using primers S-M and T-R. The sequences of primers are listed in **Supplementary file 2D**. (**D**) Schematic representation of the primers U-L1 and U-L. The sequences of primers are listed in **Supplementary file 2D**. (**E**) Genomic DNA of macrophages from transgenic and WT goat was extracted. The products of PCR were amplified using primers U-L1 and U-L. (**F**) The mRNA expression of *TLR2-4* in macrophages was detected by RT-PCR. 331 bp length fragments were amplified by Primer P1, and 188 bp

*Figure 2 continued on next page*

*Figure 2 continued*

length fragments were amplified by Primer P2. *β-actin* was used as housekeeping gene. (**G**) The protein level of TLR2-4 in macrophages was detected by Western blot using MyC-tag mouse monoclonal antibody. (**H**) The levels of TLR2-4 on macrophages membrane were analyzed by flow cytometry. Macrophages were labeled with MyC-tag monoclonal primary antibody at 4°C for 1 hr. Alexa Fluor 594 goat anti-mouse IgG was used as secondary antibody. The mean fluorescence intensity (MFI) of negative control (black), WT (blue), and TLR2-4 (red) in macrophages was estimated by FlowJo software, and then the percentage of positive TLR2-4 macrophages was calculated relative to the control sample. (**I**) The expression of TLR2-4 in macrophages was detected by immunofluorescence. Macrophages were labeled with MyC-tag monoclonal primary antibody at 4°C for 12 hr, and then goat anti-mouse secondary antibody IgG conjugated with Alexa Fluor 594-labeled was followed. The nucleus was stained with DAPI (blue). Scale bar: 100 μm.

The online version of this article includes the following source data and figure supplement(s) for figure 2:

**Source data 1.** The original gels of *Figure 2C*.

**Source data 2.** The original gels of *Figure 2E*.

**Source data 3.** The original gels of *Figure 2F*.

**Source data 4.** The original blots of *Figure 2G*.

**Figure supplement 1.** Identification of the insertion site and sequences of Toll-like receptor 2-4 (TLR2-4) in macrophages.

**Figure supplement 1—source data 1.** The original blots of *Figure 2—figure supplement 1H*.

TLR2-4 macrophage was higher than that of WT macrophage. Subsequently, clearance capacity to *S. aureus* of TLR2-4 macrophages was evaluated by colony-forming units (CFUs) test. At different time after live *S. aureus* infection, the clearance rate of *S. aureus* in TLR2-4 macrophages was significantly augmented at MOI 10 (*Figure 3C*). These results confirmed that TLR2-4 not only elevated phagocytotic capacity to *S. aureus*, but also enhanced clearance capacity in macrophages.

TLR4 can facilitate the clearance of bacteria in macrophages through the autophagy (*Liu et al., 2019*; *Pahari et al., 2020*). To investigate whether TLR2-4 induced autophagy in macrophages, the number of the autophagosomes was detected. It was higher in TLR2-4 macrophages than that in WT macrophages after *S. aureus* stimulation (MOI = 10) for 4 hr under transmission electron microscopy (TEM) (*Figure 3D*). There was more markedly distribution of puncta formation of GFP-LC3 in TLR2-4 macrophages treated with *S. aureus* (*Figure 3E*, *Figure 2—figure supplement 1G*). The conversion of LC3-I to LC3-II is one of the fundamental indicators of autophagy formation (*Cemma and Brumell, 2012*; *Kabeya et al., 2000*). We monitored the ratio between LC3-I and LC3-II by Western blot. The levels of LC3-II were up-regulated in TLR2-4 macrophages with *S. aureus* treatment at different times (*Figure 3F*, *Figure 3—source data 1*). To eliminate the inter-individual differences, fibroblast cell line produced clone goat was used to confirm the function of TLR2-4 with HK-SA stimulation. Compared to WT fibroblast cell line, the changes for levels of LC3-II were coincident with that in macrophages (*Figure 2—figure supplement 1H*). And it was found that the LC3I expression in untreated macrophages and fibroblasts were different, which suggested that TLR2-4 might affect the expression of LC3I in different cells. We further measured the conversion of LC3-I to LC3-II and the degradation of autophagy receptor SQSTM1/p62 in the presence of the autophagy inhibitor 3-methyladenine (3MA) and lysosomal inhibitor chloroquine (CQ) (*Figure 3G*, *Figure 3—source data 2*). CQ impairs autophagosome-lysosome fusion and lysosomal degradative activity. *S. aureus*-induced up-regulation of LC3-II was inhibited by autophagy inhibitor 3MA and was accumulated after CQ treatment, which proved that the conversion of LC3-I to LC3-II in macrophages is due to the enhancement of autophagy but not the impairment of lysosomal delivery of LC3-positive vacuoles. The autophagy receptor p62, an essential adaptor for target cargo, is degraded by proteolytic enzyme in the process of autophagy, thus the increase of p62 usually implicates an impairment of autophagy pathway. The down-regulation of p62 was restored by CQ treatment, suggesting the decrease of p62 in *S. aureus*-treated macrophages results from the activation of autophagy but not the impairment of delivery process of autophagosomes to lysosomes. All together, these results indicated that TLR2-4 increased the *S. aureus*-induced autophagy activity in goat macrophages. Removal of intracellular bacteria is one of autophagy function for maintaining cellular homeostasis (*Deretic, 2009*). The elimination of *S. aureus* was markedly decreased at the presence of 3MA regardless of TLR2-4 and WT macrophages (*Figure 3H*), suggesting that the autophagy was involved in clearance of *S. aureus* in TLR2-4 macrophages.

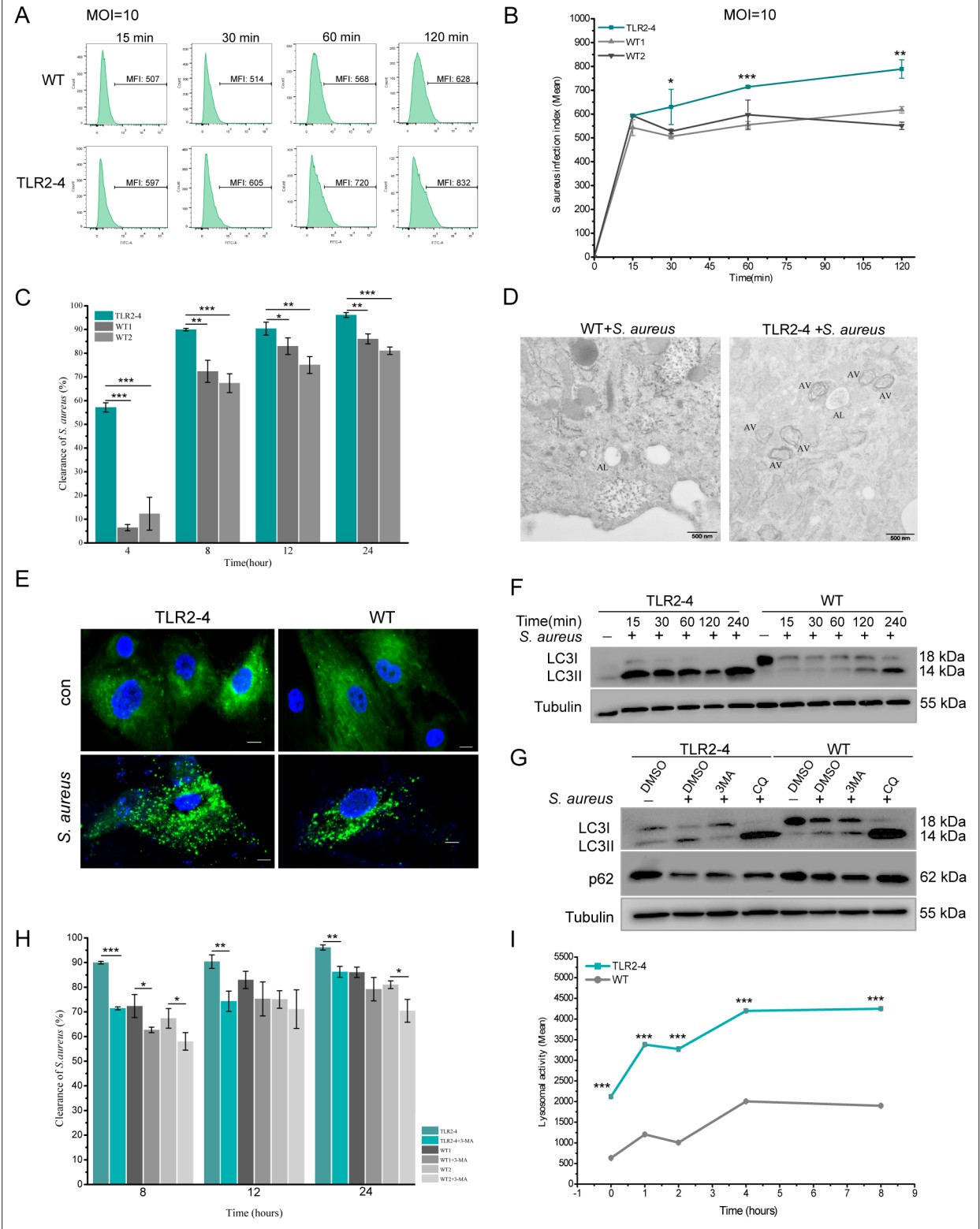

**Figure 3.** Toll-like receptor 2-4 (TLR2-4) promoted autophagy-dependent elimination of *Staphylococcus aureus* in macrophages. (**A**) *S. aureus* burden in macrophages was estimated by the mean fluorescence intensity (MFI) of macrophages infected with fluorescein isothiocyanate (FITC)-labeled *S. aureus* using flow cytometry (multiplicity of infection [MOI] = 10). MFI of FITC in macrophages was calculated by FlowJo software. (**B**) *S. aureus* infection index in macrophages was compared according to the results of MFI (n=3 biologically independent samples). (**C**) The clearance of *S. aureus* in macrophages was assessed by colony-forming unit (CFU) method. Macrophages were treated with *S. aureus* (MOI = 10) for 1 hr. After 4, 8, 12, 24 hr, survived *S.*

*Figure 3 continued on next page*

*Figure 3 continued*

*aureus* in macrophages were counted by CFU plating (n=3 biologically independent samples). (**D**) Macrophages were treated with *S. aureus* (MOI = 10) for 4 hr. *S. aureus*-induced autophagy was analyzed by TEM. AV, autophagic vacuole; AL, autophagolysosome. Scale bar: 500 nm. (**E**) LC3-punctas in macrophages were detected after *S. aureus* challenge. Macrophages were transfected with pGMLV-GFP-LC3-Puro lentivirus for 48 hr, followed by *S. aureus* treatment for 4 hr (MOI = 10). Nucleus and *S. aureus*: blue (DAPI staining); scale bar: 10 μm. (**F**) Macrophages were stimulated with *S. aureus* at the MOI of 10. The conversion of LC3I to LC3II in *S. aureus*-infected macrophages was analyzed by Western blot. (**G**) The effect of autophagy inhibitor on the level of LC3II and p62 in *S. aureus*-infected macrophages. Macrophages were pretreated with DMSO (0.1%), 3-methyladenine (3MA) (20 mM), or chloroquine (CQ) (30 μM) for 12 hr, and subsequently treated with *S. aureus* (MOI = 10) for 4 hr. (**H**) Macrophages were pretreated with either DMSO or 3MA for 12 hr, and then stimulated with *S. aureus* (MOI = 10) for 1 hr. After 8, 12, 24 hr, survived *S. aureus* in macrophages were counted by CFU plating (n=3 biologically independent samples). (**I**) The lysosomal intracellular activity of macrophages was tested by flow cytometry. Macrophages were challenged with *S. aureus* for 1 hr (MOI = 10). After washing three times with PBS, cells were cultured in DMEM medium containing gentamicin for 0, 1, 2, 4, 8 hr. Cells were collected to measure lysosomal activity by Lysosomal Intracellular Activity Assay Kit (n=3 biologically independent samples). Data are means ± SD. Student's t tests were used for comparisons between two groups. \*\*\*p<0.001, \*\*p<0.01, \*p<0.05. Exact p-values are provided in *Supplementary file 1*.

The online version of this article includes the following source data for figure 3:

**Source data 1.** The original blots of *Figure 3F*.

**Source data 2.** The original blots of *Figure 3G*.

In the process of autophagy, the 'cargo' is sequestered within autophagosomes and subsequently fused with lysosomes for digestion (*Kuo et al., 2018*; *Yu et al., 2018*). As expected, lysosomal intracellular activity was significantly higher in TLR2-4 macrophages than that in WT macrophages after *S. aureus* stimulation (*Figure 3I*). The lysosomal activity of WT macrophages was similar to that of the TLR2-4 macrophages treated with Bafilomycin A1 (*Figure 2—figure supplement 1I*). To confirm the presence of *S. aureus* in autolysosomes, GFP-LC3, lysosome, and *S. aureus* were co-located in TLR2-4 macrophages (*Figure 2—figure supplement 1J*). Taken together, these data proved that TLR2-4 promoted clearance of *S. aureus* in macrophages through the xenophagy process (the autophagy-mediated clearance of pathogens).

## *S. aureus*-induced autophagy-related genes change in the global transcriptome of TLR2-4 macrophages

To explore the mechanism of TLR2-4 regulating *autophagy*-dependent resistance against *S. aureus* in macrophages, the global transcriptomic analysis was performed by RNA-seq using TLR2-4 and two WT macrophages with *S. aureus* treatment (MOI = 10). First, according to total transcripts, WT1 was significantly correlated with WT2 by Spearsman's correlation coefficient analysis, suggesting the two WT goats are repeatability (*Figure 4A*). The individual of TLR2-4 and WT were clearly discriminated by the principal component analysis (PCA) (*Figure 4—figure supplement 1A*); 1310 DEGs between TLR2-4 and WT macrophages were obtained, 751 of which were up-regulated, whereas 559 were down-regulated (*Figure 4—figure supplement 1B*). The significant terms of DEGs were mainly involved in MAPK, cytokine activity, and TLR signaling pathway by GO enrichment analysis (*Figure 4B*). KEGG pathways analysis revealed that the significantly enriched pathways were mainly involved in NF-κB signaling pathway, cyclic adenosine phosphate (cAMP) signaling pathway, and MAPK signaling pathway (*Figure 4C*). Further, the transcripts involved in autophagy were clustered (*Figure 4D*). One-hundred and ninety-five autophagy-related DEGs were found, including the *ATG* genes, 141 and 54 of which DEGs were up-regulated and down-regulated, receptively (*Figure 4D*). Nine significant functional pathways closely related to autophagy, including endocytosis, phagosome, lysosome, and TLR signaling pathway, were enriched by KEGG analysis (*Figure 4E*). These results suggested that TLR signaling pathway, cAMP signaling pathway, and MAPK might be involved in *S. aureus*-induced autophagy in TLR2-4 macrophages.

## TLR2-4 enhanced autophagy-dependent elimination of *S. aureus* via activating JNK/ERK signaling

TIR domain of TLR4 can activate TAK1 to transduce signaling cascade (*Akira and Takeda, 2004*). MAPK signaling cascade is involved in both autophagy and TLR signaling (*Fang et al., 2014*; *Kuo et al., 2018*). The expression of MyD88 in TLR2-4 macrophages was higher than that in WT macrophages, suggesting that TLR2-4 can initiate MyD88-mediated signaling cascade (*Figure 5—figure supplement*

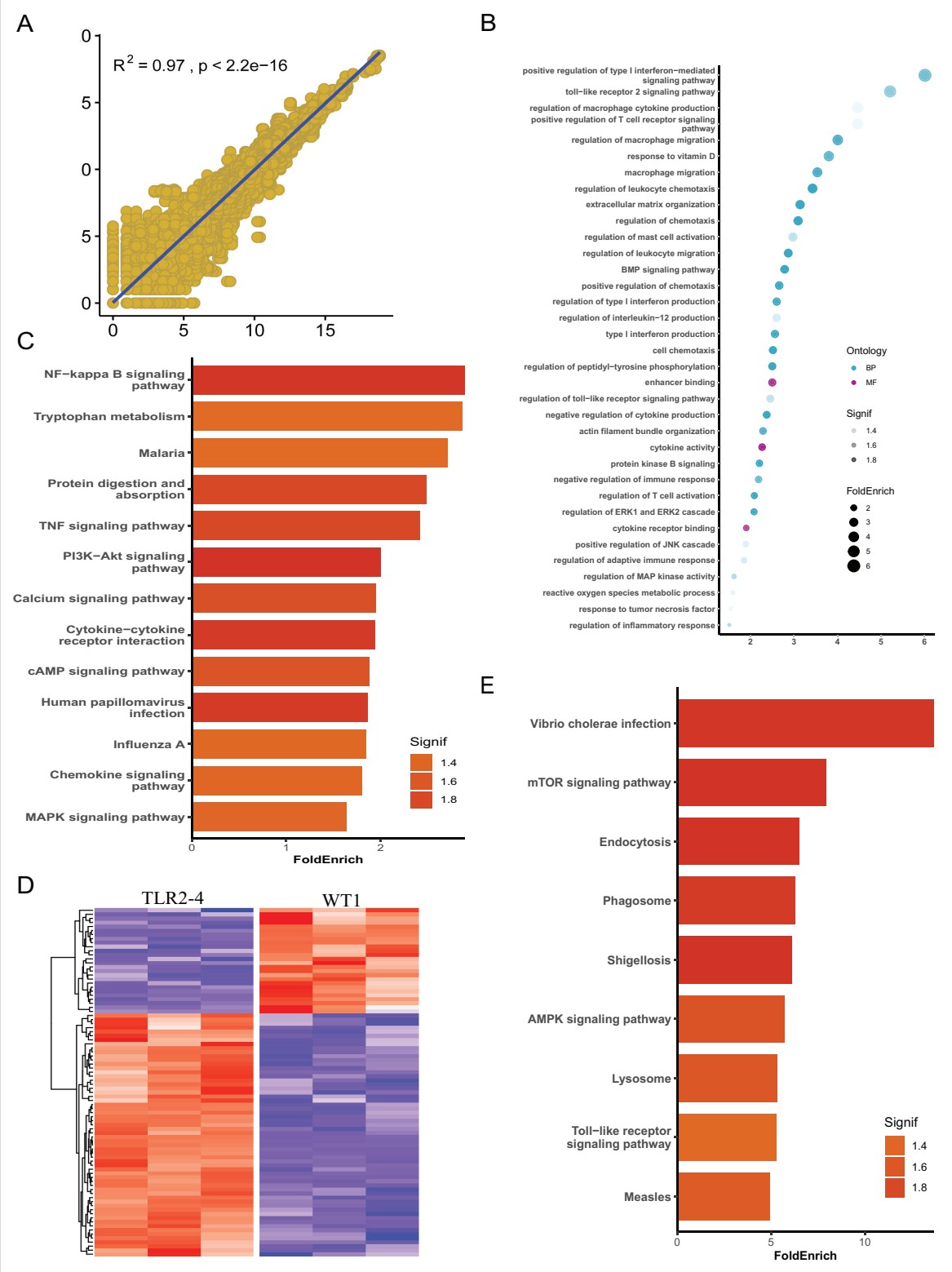

**Figure 4.** Identification of autophagy-related genes involved in clearance of *Staphylococcus aureus* in transgenic macrophages. Macrophages from the Toll-like receptor 2-4 (TLR2-4) and two wild-type goats (WT1, WT2) were infected with *S. aureus* for 1 hr (multiplicity of infection [MOI] = 10), and then after washing three times with PBS, were cultured in DMEM containing gentamicin (200 µg/ml) for 1 hr. Subsequently, total RNA of macrophages was extracted to perform RNA-seq when macrophages were cultured with DMEM containing gentamicin (25 µg/ml) for 8 hr. (**A**) Spearsman's correlation coefficient were calculated between the two wild-type macrophages on the basis of total transcripts. (**B**) Gene Ontology (GO) enrichment terms of

*Figure 4 continued on next page*

Figure 4 continued

differentially expressed genes (DEGs). (**C**) The enriched Kyoto Encyclopedia of Genes and Genomes (KEGG) pathways of DEGs. (**D**) Heatmap of 195 DEGs related to autophagy. (**E**) KEGG analysis of DEGs related to autophagy.

The online version of this article includes the following figure supplement(s) for figure 4:

**Figure supplement 1.** The principal component analysis (PCA) between two wild-type (WT) macrophages and the heatmap of the differentially expressed genes (DEGs) between TLR2-4 and WT macrophages after *S. aureus* infection.

---

*1A*). The phosphorylation of JNK and ERK, but not p38 serine-threonine kinases (p38), was triggered in both TLR2-4 and WT macrophages upon *S. aureus* treatment, and were distinctly increased in TLR2-4 macrophages after *S. aureus* stimulation compared to WT macrophages (*Figure 5A*, *Figure 5—source data 1*). That was abrogated by upstream TAK1 inhibitor Takinib (*Figure 5A*). Meanwhile, compared to WT macrophages, the conversion of LC3-I to LC3-II was prominently increased in TLR2-4 macrophages with *S. aureus* treatment, and depressed by SP600125 (JNK inhibitor) and PD98059 (ERK inhibitor) (*Figure 5B*, *Figure 5—source data 2*). These results suggested that inhibition of JNK or ERK signaling led to decrease of autophagy level induced by *S. aureus*. As shown in *Figure 5C*, the clearance of *S. aureus* was decreased more dramatically in TLR2-4 macrophages upon JNK or ERK1/2 inhibitor, indicating that TLR2-4 triggered JNK and ERK signaling to enhance *S. aureus* removal of macrophages.

TLRs can migrate from cell membrane into endosomes of the cytoplasm to activate TBK1 signaling cascade (*Akira and Takeda, 2004*; *Wei et al., 2019*). The phosphorylation of IRF3, a downstream of TBK1, was pronouncedly increased in two macrophages with *S. aureus* stimulation, and declined by endocytosis inhibitor Dynasore and TBK1 inhibitor Amlexanox (*Figure 5—figure supplement 1B*), revealing that TBK1 signaling cascade was initiated. The phosphorylation of JNK and ERK was suppressed in TLR2-4 and WT macrophages with inhibitor Amlexanox (*Figure 5A*), suggesting that TBK1 signaling promoted the activation of JNK and ERK. In addition, inhibitor Dynasore and Amlexanox also repressed the conversion of LC3-I to LC3-II (*Figure 5B*). These data demonstrated that TLR2-4 could also heighten *S. aureus*-induced autophagy via activating TBK1-JNK/ERK signaling.

The TFEB was discovered as a key transcription factor for host defense against *S. aureus* by driving the expression of autophagy genes including OPTN (*Visvikis et al., 2014*). TBK1 phosphorylated OPTN to enhance LC3 binding affinity and autophagic clearance of cytosolic bacteria (*Wild et al., 2011*). Compared with WT macrophages, there was significant aggravation of nuclear translocation of TFEB and was higher level of OPTN in TLR2-4 macrophages with *S. aureus* infection, and was pronounced decline after the inhibitor Amlexanox and Dynasore treatment (*Figure 5D*, *Figure 5—source data 3*). These results provided that the TLR2-4 could also activate the TBK1-TFEB-OPTN signaling to regulate *S. aureus*-induced autophagy.

## TLR2-4 elevated the expression of ATGs by cAMP signaling

Autophagy is mediated by conserved autophagy-related proteins at the different steps of the autophagy process. Autophagosomes are formed through the concerted action of the ATGs (*Cadwell, 2016*). Consequently, we monitored the expression of *ATG5* and *ATG12* in *S. aureus*-infected macrophages. The relative mRNA expression of *ATG5* and *ATG12* in TLR2-4 macrophages with *S. aureus* infection was significantly higher than that in the WT macrophages (*Figure 6A*), confirming that TLR2-4 enhanced the expression of autophagy-related genes. Previous studies demonstrated that the activation of JNK and ERK signaling promote the expression of *ATG5* and *ATG12* (*Wang et al., 2019*; *Xu et al., 2018*). In TLR2-4 macrophages, the expression of *ATG5* and *ATG12* was substantially increased by JNK or ERK inhibitor after *S. aureus* treatment, similar with the WT macrophages (*Figure 6B* and *Figure 5—figure supplement 1C*), suggesting that inhibition of JNK and ERK signaling increased the expression of the *ATG5* and *ATG12*. Taken together, these results demonstrated that there might be other signaling pathways regulating the *S. aureus*-induced expression of *ATG5* and *ATG12* in TLR2-4 macrophages. Forkhead box transcription factor O1 (FoxO1), a major regulator of autophagy, up-regulates the expression of ATGs (*Sengupta et al., 2009*; *Xing et al., 2018*). Its activity is tightly regulated by the phosphoinositide-3-kinase-AKT (PI3K-AKT) pathway including phosphorylation, acetylation, and ubiquitination of FoxO1 (*Fan et al., 2010*; *Nakae et al., 2000*). Compared to macrophages without *S. aureus* challenge, the phosphorylation of AKT and FoxO1 as well as acetylation of FoxO1 were hardly changed in both TLR2-4 and WT macrophages after *S. aureus* infection (*Figure 5—figure*

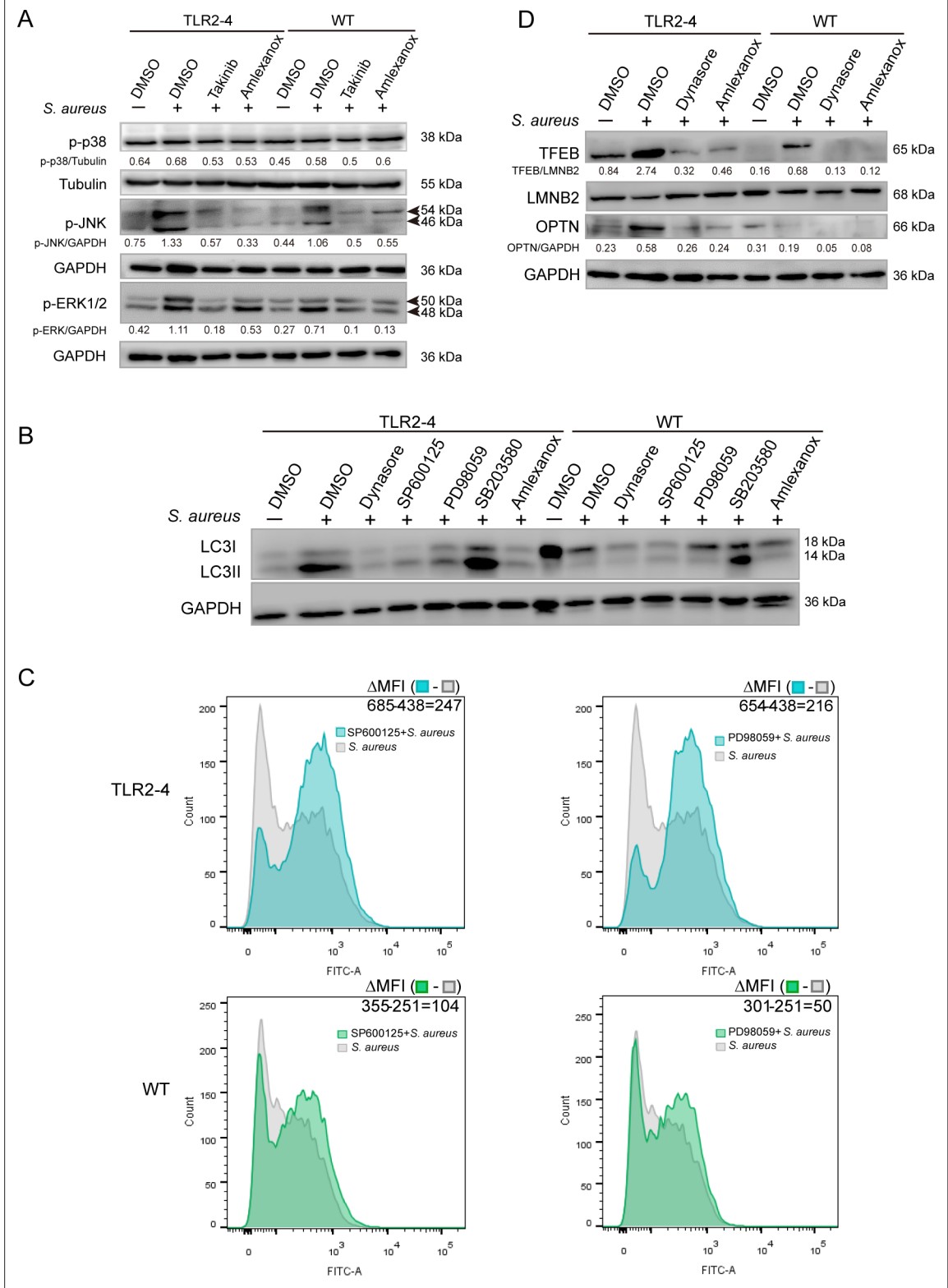

**Figure 5.** Toll-like receptor (TLR2-4) enhanced *Staphylococcus aureus*-induced autophagy via activating JNK/ERK signaling. (**A**) The change in phosphorylation of JNK, ERK, and p38 in *S. aureus*-infected macrophages. Macrophages were pretreated with DMSO (carrier), Takinib (200 µM), or Amlexanox (100 µM) for 12 hr, then stimulated with *S. aureus* for 4 hr. Phosphorylation of JNK, ERK, and p38 was observed by Western blot. (**B**) The level of LC3-II was detected in macrophages after *S. aureus* challenge. Macrophages were pretreated with DMSO, Dynasore (50 µM), SP600125 (200 µM), PD98059 (200 µM), SB203580 (200 µM), or Amlexanox for 12 hr, respectively, then treated with *S. aureus* (multiplicity of infection [MOI] = 10) for 4 hr.

*Figure 5 continued on next page*

*Figure 5 continued*

Conversion of LC3-I to LC3-II was monitored by Western blot. GAPDH protein was used as a control. (**C**) The effect activating JNK and ERK1/2 on removal of *S. aureus* in macrophages. Macrophages were treated with DMSO, SP600125, or PD98059 for 12 hr prior to stimulation with fluorescein isothiocyanate (FITC)-labeled *S. aureus* (MOI = 10) for 4 hr. The mean fluorescence intensity (MFI) of FITC in macrophages was measured by flow cytometry. (**D**) The expression of TFEB and OPTN was analyzed in macrophages by Western blot. Macrophages were pretreated with DMSO, Dynasore, or Amlexanox for 12 hr, then challenged with *S. aureus* for 4 hr (MOI = 10). The nucleoprotein was extracted and analyzed by Western blotting to detect the translocation of TFEB. The expression of OPTN was detected by Western blot. LMNB2 and GAPDH were used to normalize samples.

The online version of this article includes the following source data and figure supplement(s) for figure 5:

**Source data 1.** The original blots of *Figure 5A, B and D*.

**Source data 2.** The original blots of *Figure 5A,B,D*.

**Source data 3.** The original blots of *Figure 5A,B,D*.

**Figure supplement 1.** The phosphorylation of IRF3 was detected by Western blotting in *Staphylococcus aureus*-infected Toll-like receptor 2-4 (TLR2-4) macrophages.

**Figure supplement 1—source data 1.** The original blots of *Figure 5—figure supplement 1A*.

**Figure supplement 1—source data 2.** The original blots of *Figure 5—figure supplement 1B*.

**Figure supplement 1—source data 3.** The original blots of *Figure 5—figure supplement 1D*.

*supplement 1D*). Similar results were observed under PI3K inhibitor LY294002 treatment (*Figure 5— figure supplement 1D*). These data suggested that the PI3K-AKT-FoxO1signaling was barely involved in regulating *S. aureus*-induced autophagy in TLR2-4 macrophages.

The cAMP/PKA signaling enhanced the induction of autophagy (*Zhou et al., 2017*). Based on the results of KEGG (*Figure 4C*), we analyzed the activation of cAMP-PKA signaling in macrophages. The intracellular level of cAMP in TLR2-4 macrophages was significantly higher than that in WT macrophages regardless as *S. aureus* infection (*Figure 6C*). Simultaneously, the level of PKA catalytic subunits (PKAc) in TLR2-4 macrophages independent of *S. aureus* challenge was significantly increased than WT macrophages, and repressed by the PKA inhibitor H-89 (*Figure 6D*, *Figure 6—source data 1*). Further, the level of PKAc was extremely decreased in TLR2-4 macrophages by siRNA-mediated knockdown of TLR2-4 (*Figure 6E*, *Figure 6—source data 2*), indicating PKAc was directly regulated by TLR2-4. The cAMP-PKA signaling inhibits NF-κB activity by blocking phosphorylation of IκB (*Gerlo et al., 2011*; *Minguet et al., 2005*; *Ouchi et al., 2000*), and NF-κB suppresses the expression of *ATG5* and *ATG12* (*Lim et al., 2012*; *Shu et al., 2020*). The levels of IκB-α phosphorylation and p65 nuclear translocation in TLR2-4 macrophages were extremely reduced than that in WT macrophages regardless as *S. aureus* infection (*Figure 6D*), while inhibitor H-89 rescued the phosphorylation of IκB-α and nuclear translocation of p65 (*Figure 6D*). These results suggested that TLR2-4 inhibited the activity of NF-κB via cAMP-PKA signaling. Upon *S. aureus* stimulation, the mRNA relative expression of *ATG5* and *ATG12* in TLR2-4 macrophages was significantly increased by cAMP activator Forskolin, but was decreased by inhibitor H-89 (*Figure 6F*). These data indicated that cAMP-PKA-NF-κB signaling enhanced the expression of *ATG5* and *ATG12*. Meantime, the conversion of LC3-I to LC3-II was increased in TLR2-4 macrophages with *S. aureus* treatment by activator Forskolin, while was decreased by inhibitor H-89 (*Figure 6G*, *Figure 6—source data 3*). Taken together, TLR2-4 increased the levels of *ATG5* and *ATG12* via cAMP-PKA-NF-κB signaling to enhance *S. aureus*-induced autophagy in macrophages.

Overall, our findings suggested that domain fusion TLR2-4 enhanced autophagy-dependent elimination of *S. aureus* in goat macrophage via triggering TAK1/TBK1-JNK/ERK, TBK1-TFEB-OPTN and cAMP-PKA-NF-κB-ATGs signaling pathways (*Figure 7*).

## Discussion

It is crucial to exploit a new approach that solves the emergence of *S. aureus* infections. Modulation of host innate immunity may be a potential strategy to eliminate pathogens. TLR2 plays an essential role in recognizing *S. aureus* and activating autophagy pathway during *S. aureus* infections. However, we cannot regulate autophagy by overexpressing TLR2, as it is required to form heterodimerizes with TLR6 or TLR1. TLR4 and myeloid differentiation factor 2 complex binds to lipopolysaccharide triggers formation of the activated homodimer (*Park et al., 2009*). It is an essential component in host resistance via downstream signaling pathways initiating autophagy to degrade pathogens (*Pahari et al.,*

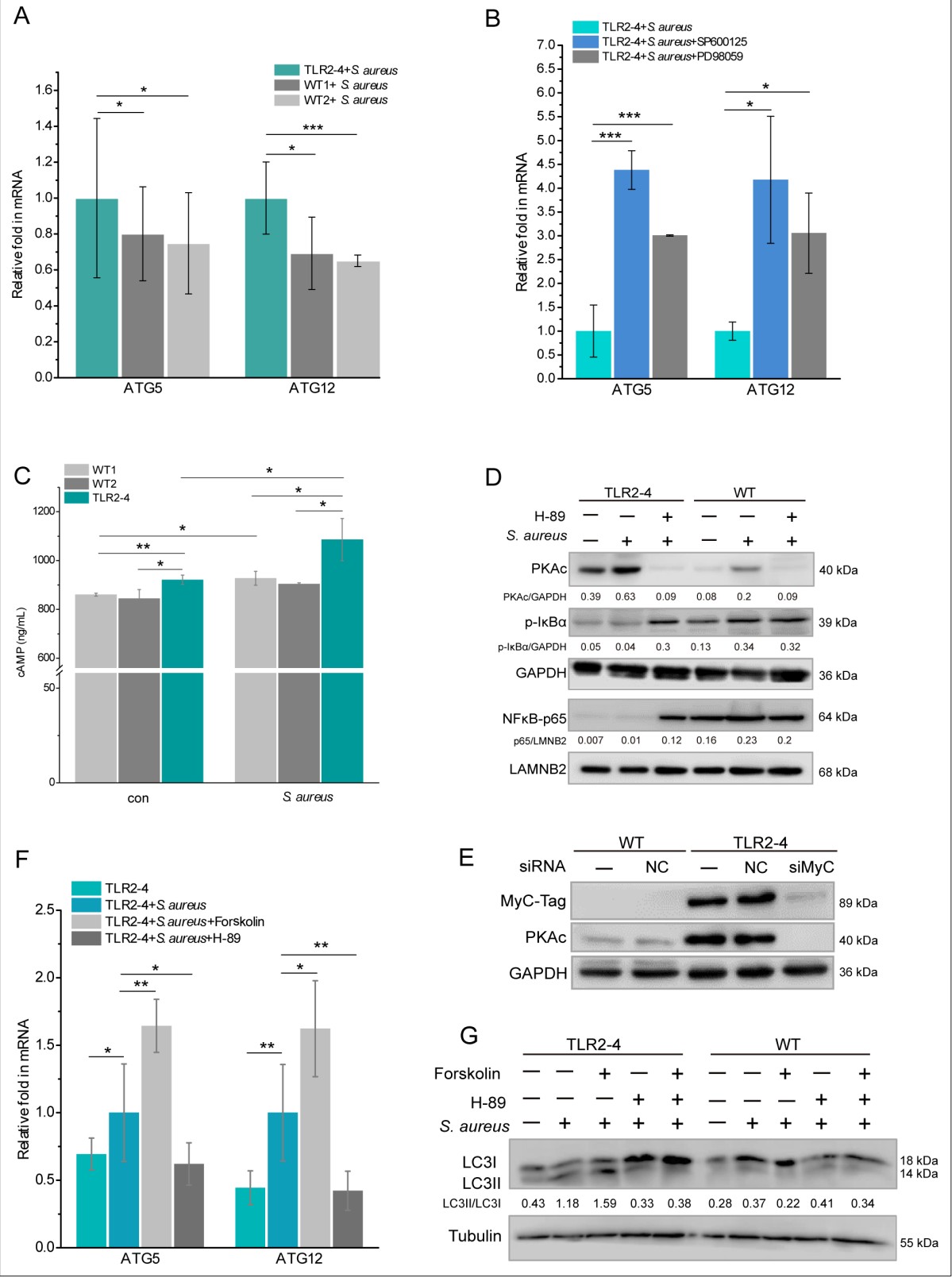

**Figure 6.** Toll-like receptor 2-4 (TLR2-4) induced autophagy flux through the cyclic adenosine phosphate (cAMP) pathway in *Staphylococcus aureus*-infected macrophages. (**A**) TLR2-4 promoted the expression of *ATG5* and *ATG12* in macrophages after *S. aureus* infection. The mRNA relative expression of *ATG5* and *ATG12* was monitored by real-time reverse transcription PCR (qRT-PCR) when macrophages were stimulated with *S. aureus* (multiplicity of infection [MOI] = 10) for 4 hr (n=3 biologically independent samples). (**B**) Inhibiting JNK or ERK1/2 increased the mRNA level of *ATG*5 and

*Figure 6 continued on next page*

*Figure 6 continued*

*ATG12* in TLR2-4 macrophages. TLR2-4 macrophages were pretreated with DMSO, SP600125 (200 mM), or PD98059 (200 mM) for 12 hr, and infected with *S. aureus* (MOI = 10) for 4 hr. qRT-PCR was performed to monitor the expression of *ATG5* and *ATG12* (n=3 biologically independent samples). (**C**) The levels of cytosolic cAMP were tested by enzyme-linked immunosorbent assay (ELISA) in macrophages. Macrophages were infected with *S. aureus* (MOI = 10) for 4 hr, and cytosolic cAMP was measured by ELISA kit. Con: un-treated cells (n=3 biologically independent samples). (**D**) The activation of PKA-NF-kB pathway was analyzed by Western blot (WB) in macrophages. After 12 hr pretreatment with either DMSO or H-89 (30 mM), macrophages were challenged for 4 hr with *S. aureus* (MOI = 10). The phosphorylation of IkB-a and the nuclear translocation of NF-kB p65 were analyzed by WB. (**E**) TLR2-4 macrophages were transfected with MyC-tag small interfering RNA (siRNA) or the negative control siRNA to knock down TLR2-4 expression. The expression of PKAc was analyzed by WB. (**F**) The effect of cAMP-PKA pathway on expression of *ATG5* and *ATG12* in TLR2-4 macrophages. Macrophages were incubated with DMSO, Forskolin (50 mM), or H-89 for 12 hr and then infected with *S. aureus* for 4 hr. The mRNA level of *ATG5* and *ATG12* were detected by qRT-PCR (n=3 biologically independent samples). (**G**) The effect of cAMP-PKA pathway on the level of LC3-II in macrophages. Macrophages were incubated with DMSO, Forskolin or/and, H-89 for 12 hr and then infected with *S. aureus* for 4 hr. The conversion of LC3-I to LC3-II were checked by WB. Data are means ± SD. Student's t tests were used for comparisons between two groups. ***p<0.001, **p<0.01, *p<0.05. Exact p-values are provided in **Supplementary file 1**.

The online version of this article includes the following source data for figure 6:

**Source data 1.** The original blots of *Figure 6D, E and G*.

**Source data 2.** The original blots of *Figure 6D,E,G*.

**Source data 3.** The original blots of *Figure 6D,E,G*.

*2020*; *Xu et al., 2007*). Meanwhile, TLR4 is involved in immune response during *S. aureus*-induced infection (*Liu et al., 2013*; *Stenzel et al., 2008*). The function of TLR2- and TLR4-dependent autophagy has been shown to be crucial in the bactericidal activity of *S. aureus* in mouse macrophages (*Shu et al., 2020*). Therefore, to achieve the enhanced host resistance against *S. aureus*, a recombinant receptor TLR2-4 with the extracellular domain of TLR2 (recognizing *S. aureus*) and the intracellular domain of TLR4 (initiating autophagy pathways) was assembled. The clone goats expressing domain fusion *TLR2-4* were generated via CRISPR/Cas9-mediated HMEJ. We first showed that the recombinant receptor TLR2-4 could recognize TLR2 ligands or *S. aureus*. Further, we demonstrated that *TLR2-4* enhance the killing of *S. aureus* by promoting autophagy pathway. This work is meaningful

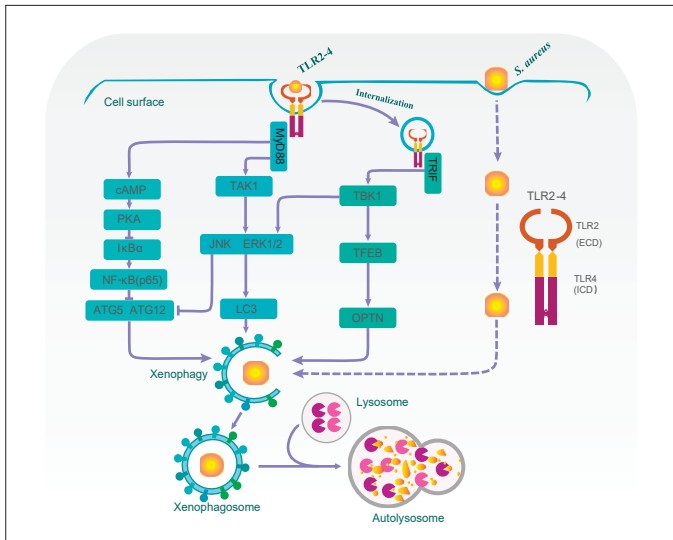

**Figure 7.** A proposed model for Toll-like receptor 2-4 (TLR2-4) enhancing autophagy-dependent clearance of *Staphylococcus aureus*. Upon *S. aureus* stimulation, TLR2-4 on the surface of macrophages augmented LC3-promoted phagophore formation to enhance xenophagy through the MyD88-dependent TAK1-JNK/ERK signaling. Internalized TLR2-4 up-regulated the autophagy receptor OPTN to promote xenophagy by activating TRIF-dependent TBK1-TFEB signaling. Meanwhile, TLR2-4 heightened the level of *ATG5* and *ATG12* to increased *S. aureus*-induced autophagy via cAMP-PKA signaling inhibiting the activity of NF-κB. Consequently, TLR2-4 enhanced the autophagy-dependent elimination of *S. aureus* in goat macrophages via three signaling pathways, including TAK1/TBK1-JNK/ERK, TBK1-TFEB-OPTN, and cAMP-PKA-NF-κB-ATGs.

for producing the *S. aureus*-resistant livestock germplasm, and help to deeply understand the mechanism on resistant to intracellular *S. aureus* infection.

The changes of cytokines are a dynamic process during immune responses. It has been reported that overexpression of TLR4 only promoted the production of proinflammatory cytokines at the early stage of LPS stimulation. After LPS treatment for more than 4 hr, the TLR4-mediated TRIF-dependent pathway was activated, and the cytokines secreted by TLR4-overexpressing macrophages were reduced compared with WT macrophages (*Wei et al., 2019*). As previously reported, the mRNA level of IL-6 in TLR2-4 fibroblast cells with siRNA TLR2 was dramatically increased, while IL-8 and IL-1β were significantly decreased after stimulated by TLR2 ligands for 10 hr. Therefore, we hypothesized that the activation of IL-8 and IL-1β mediated by TLR2-4 may be inhibited through TRIF-induced internalization pathway. In addition, it has been reported that pretreatment of microglia with P3C induces the secretion of anti-inflammatory cytokine and attenuates inflammatory response (*Kochan et al., 2012*). These results demonstrated that TLR2-4 recognizes ligands by the extracellular domain of TLR2 and initiates downstream immune response by the intracellular domain of TLR4. P3C is a synthetic triacylated lipopeptide and a TLR2/TLR1 ligand. P2C is a synthetic diacylated lipopeptide and is recognized by the TLR2/TLR6 heterodimer. They mimic the acylated amino terminus of bacterial lipopeptides and are potent activator of the proinflammatory transcription factor NF-κB and AP-1. However, *S. aureus* is the whole cell pathogen with PAMPs on surface that include lipoteichoic acid (LTA) and peptidoglycan (PGN) in addition to lipopeptides. LTA and PGN have been reported to induce the release of IL-8 through MyD88-dependent signaling pathway (*Hattar et al., 2017*; *Zenhom et al., 2012*). Therefore, the mRNA levels of downstream cytokines were different after fibroblasts stimulated by TLR ligands or *S. aureus*.

The autophagic response promotes *S. aureus* survival in zebrafish neutrophils (*Prajsnar et al., 2021*). Nevertheless, it is currently unknown whether autophagy pathway of goat macrophages is conductive to bacterial replication during *S. aureus* infection. In contrast to this publication, we observe that the autophagic response reinforced *S. aureus* clearance in goat *TLR2-4* macrophages. The elimination of *S. aureus* was significantly impaired by autophagy inhibitor 3-MA in *TLR2-4* macrophages. These data provide evidences that domain fusion *TLR2-4* maintained the ability of activating autophagy pathways to degraded pathogens and recognized *S. aureus* and its ligands.

We subsequently focused on the pathways of inducing autophagy in *S. aureus*-infected TLR2-4 macrophages. Previous work suggested that OPTN was phosphorylated by TBK1 and recruited to ubiquitinated cytosolic bacteria to enhance autophagic degradation of *Salmonella* (*Wild et al., 2011*), and TFEB regulated OPTN-induced autophagy activation in host defense against *S. aureus* (*Visvikis et al., 2014*). We suspected that the same pathway might be involved in TLR2-4 signal. The effects of inhibited TBK1 on *S. aureus*-induced autophagy in TLR2-4 macrophages were estimated. The data revealed that inhibition of TBK1 led to suppression of TFEB nuclear translocation and reduction of OPTN expression, and then diminished *S. aureus*-induced autophagy. These data suggested that the TLR2-4 was involved in TBK1-TFEB-OPTN signaling pathway in *S. aureus*-induced autophagy in macrophages.

TAK1 can be activated through the TLR4-mediated MyD88-dependent pathway on cell surface. Activated TAK1 initiates MAPK signaling cascades. In the meantime, TLR4-induced TRIF-dependent pathway triggers TBK1 in the cytoplasm, then phosphorylates JNK or ERK (*Akira and Takeda, 2004*). Our observation evidenced that TAK1 and TBK1 signal were initiated in *S. aureus*-infected macrophages expressing TLR2-4. Subsequently, the phosphorylated JNK and ERK suggested that TLR2-4 is similar to TLR4, can activate the TAK1/TBK1-JNK/ERK signaling pathways. We further show that the inhibition of JNK or ERK exactly abrogated the LC3-puncta and declined the clearance of *S. aureus*, indicating that TLR2-4-triggered JNK and ERK signal promote host resistant against *S. aureus* through autophagy pathway. These results are consistent with previous observations that JNK and ERK participate in mediating autophagy activation in pathogen-infected macrophages (*Fang et al., 2014*; *Lu et al., 2017*).

Notably, we observed that the expression of autophagy-related *ATG5* and *ATG12* were significantly augmented in *S. aureus*-infected TLR2-4 macrophages. However, inhibition of JNK or ERK significantly up-regulated the expression of ATG5 and ATG12 in TLR2-4 macrophages. These data suggest that JNK or ERK signaling enhances the expression of ATG8 family protein LC3II in *S. aureus*-infected goat macrophages, but not ATG5 and ATG12. Interestingly, the negative regulation of the

JNK signaling pathway suppresses the ATG5 and ATG12 expression in hippocampus of epileptic rats (*Wang et al., 2019*), and the ERK promotes the expression of ATG5 in the human colorectal cancer cell lines (*Xu et al., 2018*). Combining these evidences with our data indicates the MAPK modulating the expression of ATGs probably is different because of different species or cells during autophagy events. Therefore, we inferred that other signaling pathways were involved in increasing transcript of *ATG5* and *ATG12* during *S. aureus*-induced autophagic process in TLR2-4 macrophages. Previous studies showed the PI3K-AKT-FoxO1 pathway regulated the expression of ATGs in mouse ovarian granulosa cells (*Shen et al., 2017*). In addition, the transcriptomic GO and KEGG enrichment analysis demonstrated that PI3K-AKT pathway was implicated in *S. aureus*-induced autophagy. Thus, we detected the activation of AKT and FoxO1 by Western blot. The results showed that both AKT and FoxO1 were scarcely activated in *S. aureus*-infected macrophages, indicating the transcript of ATGs was not triggered by PI3K-AKT-FoxO1 signaling pathway.

Since the cAMP signal was enriched during *S. aureus*-induced autophagy in TLR2-4 macrophages by RNA-seq, we focused on the effect of cAMP signal on the expression of *ATG5* and *ATG12*. The level of cAMP in TLR2-4 macrophages was significantly higher than that in WT macrophages after *S. aureus* infection. Furthermore, increased cAMP activated the phosphorylation of PKA and inhibited NF-κB p65 nuclear translocation by disturbing the IκB-α phosphorylation. These data were similar to the previous evidences that cAMP-induced PKA down-regulated the expression of NF-κB (*Zuo et al., 2016*). NF-κB signal pathway could suppress autophagy by inhibiting the expression of *ATGs* (; *Liu et al., 2020*). We next verified LC3-II activated by cAMP-PKA-NF-κB signaling and the expression of *ATG5* and *ATG12*s in TLR2-4 macrophages using exerting the cAMP activator (Forskolin) and the PKA inhibitor (H-89). The inhibitor H-89 not only augmented the nuclear translocation of NF-κB p65, but also suppressed the expression of *ATG5* and *ATG12* in TLR2-4 macrophages after *S. aureus* challenge, while the activator Forskolin facilitated substantially the mRNA levels of *ATG5* and *ATG12*. Meanwhile, the levels of LC3II were decreased in *S. aureus*-infected TLR2-4 macrophages by the inhibitor H-89. These data suggested that TLR2-4 triggered cAMP-PKA-NF-κB signaling to increase the expression of *ATG5* and *ATG12*, and then promoted autophagic process to complete the elimination of pathogens within autophagosomes (*Keller et al., 2020*; *Mestre and Colombo, 2012*).

In summary, we first assembled the domain fusion receptor TLR2-4 and generated the clone goat expressing TLR2-4. The functions of TLR2-4 for resistant against *S. aureus* infection were elucidated in macrophages. TLR2-4 enhanced the autophagy level to effectively eliminate *S. aureus* in macrophages via three signaling pathways including TAK1/TBK1-JNK/ERK, TBK1-TFEB-OPTN, and cAMP-PKA-NF-κB-ATGs (*Figure 7*), which provided a new insight into *S. aureus* defense in dairy goat.

# Materials and methods

## Key resources table

| Reagent type (species) or resource | Designation | Source or reference | Identifiers | Additional information |
|---|---|---|---|---|
| Gene (*Capra hircus*) | TLR2 | GenBank | NC_030824.1 | |
| Gene (*Capra hircus*) | TLR4 | GenBank | NC_030815.1 | |
| Strain, strain background (*Staphylococcus aureus*) | ATCC29213 | GenBank | U77328 | Bacteria cells |
| Transfected construct | PX458 (pSpCas9(BB)–2A-GFP) | Addgene | RRID: Addgene_48138 | PMID:24157548 |
| Transfected construct | pRosa26-promoter | Addgene | RRID: Addgene_21710 | PMID:9108056 |
| Transfected construct (*Capra hircus*) | siRNA to endogenous TLR2 | GenePharma | Sense: 5′-GCACUUCAACCCUCCCUUUTT-3′ | Antisense: 5′-AAAGGGAGGGUU GAAGUGCTT-3′ |
| Sequence-based reagent | siRNA: nontargetin control | GenePharma | | Silencer Select |
| Antibody | nti-MyC (Mouse monoclonal) | Proteintech | Cat# 60003–2-lg RRID: AB_2883088 | IF (1:100), WB (1:1000) |

*Continued on next page*

*Continued*

| Reagent type (species) or resource | Designation | Source or reference | Identifiers | Additional information |
|---|---|---|---|---|
| Antibody | Anti-NF-kB p65 antibody (Rabbit polyclonal) | Abcam | Cat# ab16502 RRID: AB_443394 | WB (1:1000) |
| Antibody | Anti-LMNB2 (Rabbit polyclonal) | Beyotime | Cat#: AF0219 | WB (1:1000) |
| Antibody | Phospho-p44/42 MAPK (Rabbit polyclonal) | CST | Cat#: 9,101 RRID: AB_331646 | WB (1:1000) |
| Antibody | Phospho-JNK1/2/3-T183/T183/T221 (Rabbit monoclonal) | ABclonal | Cat#: AP0631 RRID: AB_2771232 | WB (1:1000) |
| Antibody | Tubulin (Mouse monoclonal) | Beyotime | Cat#: AT819 | WB (1:1000) |
| Antibody | Anti-LC3 (Rabbit polyclonal) | Proteintech | Cat#: 14600–1-AP RRID: AB_2137737 | WB (1:1000) |
| Antibody | Anti-SQSTM1/p62 (Rabbit polyclonal) | Affinity | Cat#: AF5384 RRID: AB_2837869 | WB (1:1000) |
| Antibody | Phospho-p38 MAPK-T180 (Rabbit polyclonal) | ABclonal | Cat#: AP0238 RRID: AB_2771307 | WB (1:1000) |
| Antibody | Anti-GAPDH (Rabbit polyclonal) | Sangon Biotech | Cat#: D110016 RRID: AB_2904600 | WB (1:1000) |
| Antibody | Anti-TFEB (Rabbit polyclonal) | Proteintech | Cat#: 13372–1-AP RRID: AB_2199611 | WB (1:1000) |
| Antibody | Anti-Optineurin (Rabbit polyclonal) | Affinity | Cat#: DF6655 RRID: AB_2838617 | WB (1:1000) |
| Antibody | Phospho-pan-AKT1/2/3 (Rabbit polyclonal) | Affinity | Cat#: AF0908 RRID: AB_2834079 | WB (1:1000) |
| Antibody | Phospho-FOXO1A (Rabbit polyclonal) | Affinity | Cat#: AF3416 RRID: AB_2834858 | WB (1:1000) |
| Antibody | Acetyl-FOXO1A (Rabbit polyclonal) | Affinity | Cat#: AF2305 RRID: AB_2845319 | WB (1:1000) |
| Antibody | Anti-PRKACA (Rabbit polyclonal) | ABclonal | Cat#: A0798 RRID: AB_2757400 | WB (1:1000) |
| Antibody | Phospho-IKB alpha (Rabbit monoclonal) | CST | Cat#: 2,859 RRID: AB_561111 | WB (1:1000) |
| Antibody | Phospho-IRF3-S386 (Rabbit polyclonal) | ABclonal | Cat#: AP0857 RRID: AB_2771209 | WB (1:1000) |
| Antibody | Peroxidase-Conjugated Goat anti-Rabbit IgG | ZSGB-BIO | Cat# ZB-2301 RRID: AB_2747412 | WB (1:5000) |
| Antibody | Peroxidase-Conjugated Goat anti-Mouse IgG | ZSGB-BIO | Cat# ZB-2305 RRID: AB_2747415 | WB (1:5000) |
| Antibody | Alexa Fluor 594 goat anti-mouse IgG | ThermoFisher Scientific | Cat#: A11005 RRID: AB_2534073 | IF (1 µg/ml) |
| Commercial assay or kit | Nuclear Extraction Kits for Cells | Invent Biotechnologies | Cat#: SC-003 | |
| Chemical compound, drug | Dynasore | MCE | Cat#: HY-15304 | 50 µM |
| Chemical compound, drug | SP600125 | MCE | Cat#: HY-12041 | 200 µM |
| Chemical compound, drug | PD98059 | MCE | Cat#: HY-12028 | 200 µM |
| Chemical compound, drug | Takinib | MCE | Cat#: HY-103490 | 200 µM |

*Continued on next page*

*Continued*

| Reagent type (species) or resource | Designation | Source or reference | Identifiers | Additional information |
|---|---|---|---|---|
| Chemical compound, drug | Amlexanox | MCE | Cat#: HY-B0713 | 100 µM |
| Software | ImageJ | http://imagej.nih.gov/ij/ | RRID: SCR_003070 | |
| Software | Flowjo | https://www.flowjo.com/solutions/flowjo | RRID: SCR_008520 | |
| Software | Origin | http://www.originlab.com/index.aspx?go=PRODUCTS/Origin | RRID: SCR_014212 | |
| Other | DAPI stain | Solarbio | Cat#: C0065 | 10 µg/ml |

## Cell culture

Goat fetal fibroblasts were cultured in DMEM/F12 supplemented with 10% fetal bovine serum (FBS, Gibco) and 1% penicillin-streptomycin. Monocyte-derived macrophages (MDMs) from goat peripheral blood were cultured in DMEM supplemented with 10% FBS and 1% penicillin-streptomycin.

## Bacterial strains and growth conditions

The *S. aureus* strain ATCC29213 were preserved in our laboratory. Bacteria were grown in tryptic soy broth (TSB) liquid medium with shaking overnight at 37°C. Cultures were diluted with TSB at a ratio of 1:100 and incubated with shaking at 37°C until the optical density at 600 nm was 1.5.

## Animals

Healthy dairy goats of Lao shan and Nubian were selected. All animal experiments and treatments were approved and supervised by the Animal Welfare Committee of China Agricultural University (Approval ID: AW31501202-1-1).

## Construction of the domain fusion TLR2-4

Goat blood RNA was extracted by blood total RNA extraction kit (TIANGEN, China) according to the manufacturer's instructions. cDNA synthesis was executed with 1 µg of total RNA with *Bca*BEST RNA PCR Kit (TIANGEN, China) as per the manufacturer's instructions. Primers TLR2-U, TLR2-L, TLR4-U, TLR4-L were used to amplify extracellular domain of goat *TLR2* and transmembrane and intracellular domain of *TLR4*. The PCR was as follows: 94°C for 2 min, followed by 35 cycles of 98°C for 10 s, 62°C for 30 s, 68°C for 60 s (extracellular region of *TLR2*) or 20 s (transmembrane and intracellular region of *TLR4*), and 68°C for 5 min. Signal peptide and MyC sequence were added to the 5' end of the *TLR2* (primer U-110), and 20 bp sequence of TLR4 transmembrane region homology sequence (primer L-41) was added to the 3' end of the *TLR2*. Primers T2-U20 and T4-L20 were used for overlap PCR to obtain *TLR2-4*.

## Screening for sgRNA targets

In the first intron of *SETD5*, the protospacer adjacent motif (PAM) was searched on the double-stranded DNA in this region, and the 20-nucleotide (nt) at its 5' end was used as the target sequence. The 13-nt seed regions at the 5' end of PAM were compared in NCBI online database to estimate the off-target effect. The website http://unafold.rna.albany.edu/q=mfold/RNA-Folding-Form/ was used to evaluate the sgRNA structure and free energy. The PX458 plasmid (Addgene: 48138) was digested with BbsI endonuclease at 37°C for 1 hr, then the vector was purified. Eight pairs of CRISPR RNAs (crRNA)-oligo primers were synthesized (***Supplementary file 2B***). CrRNAs were annealed with conditions of 95°C for 2 min, followed by dropping to 25°C at the rate of 5°C/s. The annealed oligos were diluted by adding 1 µl oligo to 99 µl ddH$_2$O. The reactions of crRNA and digested PX458 vector were performed in a final volume of 10 µl and the solution mix consisted of 1 µl diluted crRNA-oligo, 100 ng digested PX458, 1 µl T4 ligase buffer, 1 µl T4 ligase, and ddH$_2$O adding to 10 µl for the overnight at 16°C.

T7E1 assay was used in evaluation of genome editing efficiency. Cells were cultured for 3 days after transfection, and the genomes were extracted. The T7E1 primers listed in ***Supplementary file 2C***

were used for nested PCR. The PCR products were purified and annealed. The hybridized products were digested with T7E1 (NEB, Ipswich, MA) at 37°C for 30 min and separated by agarose gel. The intensity of the bands was analyzed by using ImageJ software. The indel rate $= 1 - \sqrt{\left(1 - \frac{b+c}{a+b+c}\right)}$, where $b$ and $c$ are the intensities of digestion products by T7E1 and $a$ is the intensity of undigested band.

## Construction of TLR2-4 donor vector (pRosa26-TLR2-4)

The poly(A) sequence was cloned into pRosa26-promoter vector (Addgene: 21710) by BamHI-HF and XbaI. The homologous arms were amplified by PCR (primer HA-R-U, primer HA-R-L, HA-L-U, HA-L-L, *Supplementary file 2D*). Homologous arm HA-R containing sgRNA sequence was inserted into the 3' end of poly(A) of constructed vector using XbaI and SacI-HF, and then digested with Acc65I and SalI to insert homologous arm HA-L containing sgRNA sequence into the upstream of pRosa26-promoter. The *TLR2-4* was inserted between pRosa26-promoter and poly(A) within above vector linearized by PmlI and EcoRI-HF. Finally, the constructed vector was digested with SacI-HF, and then the expression construction of CMV-tdTomato was amplified from pCMV-tdTomato vector (Clontech, 632534) to inserted downstream of HA-R by GeneArt Gibson Assembly (Thermo Fisher, A46624).

## Screening of fibroblasts expressing TLR2-4

The pRosa26-TLR2-4 and PX458-sgRNA vector were co-transfected into goat fibroblasts. After 3 days of culture, cells with both green and red fluorescence were obtained by fluorescence-activated cell sorting (FACS). Selected cells were seeded in a 10 cm cell culture dish and cultured in DMEM/F12 supplemented with 20% FBS to obtain the monoclonal cells. Cell clones can be formed after cultured for 14 days (fluorescence of monoclonal cells would disappear in days 7–10), and then clones of cells were transferred to 96-well plates. Some of the cells were selected into a new well at 70% cell confluence and the others were identified by PCR to determine whether *TLR2-4* was inserted into the *SETD5* locus using primers R-U and R-L listed in *Supplementary file 2D*.

## Western blotting

$1 \times 10^6$ cells were seeded in a 10 cm culture dish and cultured for 12 hr, and then cells were lysed in RIPA lysis buffer (Aidlab, PP1201) containing protease inhibitor cocktails, phosphatase inhibitor complex, and phenylmethylsulfonyl fluoride. The nuclear protein was extracted using a nuclear extraction kit (Invent Biotechnologies, SC-003). Briefly, after incubation on ice for 30 min, the cell lysates were prepared by centrifuge at 4°C, 14,000 rpm for 15 min. Protein concentrations determined by a bicinchoninic acid (BCA) assay were mixed with 6× Loading buffer (Beyotime Biotechnology, P0015F), and heated at 99°C for 5 min. Samples were electrophoretic separated on SDS-PAGE gels of sodium dodecyl sulfate-polyacrylamide and transferred onto polyvinylidene fluoride membranes (Millipore, Burlington, MA) at 320 mA for 60 min. The membranes were blocked for 90 min in TBST (0.5% Tween-20) with 5% non-fat powdered milk. After washing three times with TBST, the membranes were incubated overnight with primary antibodies against the proteins of interest at 4°C. Membranes were washed three times with TBST and incubated with secondary antibodies for 1 hr. Protein bands were detected using BeyoECL Star Chemiluminescent Substrate (Beyotime Biotechnology, P0018FS) and blots were digitally imaged on a Bioanalytical Imaging System (Azure Biosystems, Dublin, CA). The density of the bands was analyzed by using ImageJ software.

## Laser scanning confocal microscopy

Cells were seeded at $1 \times 10^5$ per well in a glass chamber slide (Millipore, PEZGS0816) and cultured for 12 hr. After washing twice, cells were fixed with ice methanol for 30 min, and then blocked with blocking buffer (Beyotime Biotechnology, P0102) for 1 hr. Cells were incubated with MyC-tag mouse monoclonal antibody (1:50, Proteintech, 60003-2-Ig) in staining buffer for 12 hr. Cells were washed three times with PBS, followed by incubations with Alexa Fluor 594 goat anti-mouse IgG secondary antibody (Thermo Fisher, A11005) for 1 hr. After washing with PBS for three times, cells were incubated with DAPI solution for 10 min, and the images were observed by the confocal microscope (Nikon A1, Japan).

In analysis of autophagic flux experiment, pGMLV-GFP-LC3-Puro lentivirus was used to track LC3. Macrophages were transfected with lentivirus for 48 hr following the manufacturer's instruction and then treated with *S. aureus* for 4 hr. Then the images were acquired using confocal microscopy. In co-localization experiment, Lyso-Tracker Red (Solarbio, L8010) was used as the marker of lysosome. Macrophages stably expressing GFP-LC3 were treated with *S. aureus* for 4 hr, and then cells were washed for three times and incubated with Lyso-Tracker Red working solution in 37°C incubator for 30 min. Cells were incubated with DAPI for 10 min, and then were washed three times with PBS. The images were captured by confocal microscope.

## Transfection of siRNA in goat fibroblasts

The siRNA targeting endogenous *TLR2* was designed based on the intracellular domain of TLR2, which did not affect TLR2-4 after transfection. Goat fibroblasts ($3 \times 10^5$/well) were seeded into six-well plate and cultured for 12 hr. Cells were transfected with 75 nmol/l siRNA targeting TLR2 (Gene-Pharma, China) or siRNA negative control using 0.75 µl of Lipofectamin 3000 (ThermoFisher Scientific, Waltham, MA). After 72 hr, the cells were harvested and detected the targeting efficiency at mRNA level by real-time reverse transcription PCR (qRT-PCR). The most effective siRNA was selected for the subsequent experiments.

## Generation of TLR2-4 transgenic goats

Nubian goats about 40 kg weight were selected and treated with progesterone using a controlled internal drug release (CIDR) equipment containing 300 mg progesterone. Donor goats were injected with 240 IU of follicle stimulating hormone at the time of 60 hr before CIDR was removed, and injected intramuscularly with 0.1 mg of prostaglandin at the time of CIDR removal. One-hundred IU luteinizing hormone was injected intramuscularly 38 hr after CIDR withdrawal to induce ovulation in goats. The CIDR in recipients were removed when the donors' CIDR were withdrawn. The recipients were injected with 250 IU pregnant mare serum gonadotropin.

All goats were fasted 12 hr before oocyte collection for facilitating surgery. The ovulation response was verified by laparoscopy. Goats displayed corpora lutea were selected for oocyte collection at 62 hr after CIDR removal. The oocytes were collected by oviduct flushing with 20 ml warm PBS containing 0.3% bovine serum albumin. Collected oocytes were transferred into the holding medium using an inverted microscope (LX71 Olympus, Japan).

The oocytes with cumulus cells were incubated in M199 medium (Gibco, Waltham, MA) containing 0.1% hyaluronidase for 5 min to remove cumulus cells. The nucleus of oocytes was removed by Micromanipulation system (Eppendorf, Germany), and TLR2-4 cells were injected into the perivitelline space of enucleated oocytes. Reconstructed embryos were fused by two direct current pulses of 20 V for 20 µs using an ECM2001 Electrocell Manipulator (BTX, Holliston, MA) and then were incubated in M199 supplemented with 10% FBS and 7.5 µg/ml CB at 37°C in a 5% $CO_2$ incubator to observe for 2 hr. The fused embryos were treated with 5 µmol/l ionomycin for 4 min and then incubated in mSOF medium (Sigma, St. Louis, MO, USA) containing 2 mmol/l 6-dimethylaminopurine for 3 hr. After washing for three times, embryos were transferred to mSOF medium and cultured overnight. The fused embryos were transplanted into the oviduct of recipients. The pregnancy status of the recipient was evaluated by ultrasonography at 60 days after embryo transfer.

## Isolation of macrophages from goat peripheral blood

Macrophages isolation was performed as described previously (*Wei et al., 2019*). Ten ml blood was collected from the jugular vein of TLR2-4 transgenic goat. MDMs were obtained from the peripheral blood using lymphocyte separation medium (TBD, LTS1087) according to the manufacturer's instructions. Cells were incubated at 37°C in a 5% $CO_2$ incubator for 2 hr, and the non-adherent cells were removed by washing three times with PBS. The adherent cells were cultured in DMEM (ThermoFisher Scientific, C11995500BT) containing 10% FBS. The medium was changed every 24 hr, and non-adherent cells were washed off. After 72 hr, adherent cells were mainly composed of MDMs (*Deng et al., 2017*).

## Identification of the macrophages expressing TLR2-4 by PCR

Genomic DNA of transgenic goat macrophages was extracted by HiPure Universal DNA kit (Magen, China). The genome integration of *TLR2-4* was confirmed by PCR using primer U, primer L1, and primer L (*Supplementary file 2D*). The 1188 bp exogenous fragment was only existed in transgenic macrophages by primers U-L1. The 3795 bp exogenous fragment and the 312 bp endogenous fragment were existed in TLR2-4 and WT macrophages, respectively, by primers U-L. The DNA fragment of 3795 bp was further purified by Sanger sequencing.

## Assay of the expression of TLR2-4 in macrophages by RT-PCR

Total RNA was extracted by RNApure Total RNA Kit (Aidlab, China) in accordance with its protocol. Quantity and quality of RNA were assessed using a NanoDrop 2000 spectrophotometer (Thermo Scientific, Waltham, MA). cDNAs were synthesized using HiScript 1st Strand cDNA synthesis kit (Vazyme Biotech,Nanjing, China). The PCR cycle conditions were 94°C for 3 min, followed by 35 cycles of 94°C for 30 s, 57°C for 30 s, and 72°C for 30 s, and final extension at 72°C for 5 min. The primers P1 and P2 are listed in *Supplementary file 2D*.

## Cell transfection

Approximately $2 \times 10^6$ of fibroblasts were cultured for 16 hr in DMEM/F12, and then washed twice with PBS and collected to a 1.5 ml tube for centrifugation at 1500 rpm/min for 5 min. Cells were suspended with 100 µl electrotransfer solution with 4 µg vector, and then transferred by the N-024 program of Amaxa's Nucleofector electrotometer (Lonza, Switzerland) according to the manufacturer's instructions. The transfected cells were seeded into six-well plates and cultured in DMEM/F12 media containing 20% FBS for at 37°C with 5% $CO_2$.

qRT-PCR qPCRs were prepared with SYBR Green PCR kit (QIAGEN, 208054) following the manufacturer's instructions. The reaction mixture was comprised of 10 µl of SYBR Green, 1 µl cDNA, 0.8 µl (10 µM) of forward and reverse primers and 7.4 µl RNase free water. The data normalization was performed with *β-actin*. Amplifications were performed starting with a 2 min template denaturation step at 95°C, followed by 40 cycles of 95°C for 5 s and 60°C for 10 s. Melt curves for each gene were recorded at the end of each cycle. The qPCRs were run on an Agilent StrataGene Mx3000P instrument (Agilent Technologies, Santa Clara, CA). Relative expression of genes was analyzed by the $2^{-\Delta\Delta Ct}$ method (*Livak and Schmittgen, 2001*; *Schmittgen and Livak, 2008*). All primers used in qRT-PCR are listed in *Supplementary file 2E*.

## Flow cytometry

Cells ($3 \times 10^5$/well) were seeded into the six-well plates and cultured for 12–16 hr. Cells were collected and blocked with flow staining buffer (5% FBS in PBS) for 10 min at room temperature. Cells were incubated with MyC-Tag mouse monoclonal antibody (1:500) in flow staining buffer for 1 hr at 4°C, followed by washing twice with PBS for 5 min. Incubated cells with Alexa Fluor 594-labeled goat anti-mouse IgG(H+L) (1:500) in PBS for 1 hr at 4°C, and finally washed twice with PBS for 5 min. Cells were collected and fixed with 4% paraformaldehyde for 30 min. Cells were washed twice and then resuspended in the flow staining buffer, and mediately analyzed by Flow Cytometry BD LSRFortessa (BD Company, Franklin Lakes, NJ).

## Phagocytose of *S. aureus* in macrophages

*S. aureus* was stained by fluorescein isothiocyanate (FITC) as described earlier (*Cantinieaux et al., 1989*). Macrophages were plated at a density of $3 \times 10^5$/well and infected with FITC-labeled *S. aureus* at an MOI of 1:1 or 1:10 for 15, 30, 60, 120 min, and then were washed three times with PBS. Macrophages were treated with DMEM containing gentamicin (200 µg/ml) for 1 hr to remove extracellular *S. aureus*. Cells were collected, and then fixed with 4% paraformaldehyde. The mean fluorescence intensity (MFI) of FITC-fluorescence in macrophages was detected by flow cytometry. The data was analyzed with FlowJo software.

## Clearance of *S. aureus* in macrophages

Macrophages were infected with *S. aureus* at an MOI of 1:10 for 1 hr, then washed three times with PBS. Cells were treated with DMEM containing gentamicin (200 µg/ml) for 1 hr to remove extracellular

*S. aureus*. After washing three times with PBS, cells were cultured with DMEM containing gentamicin (25 μg/ml) for 4, 8, 12, 24 hr. Cells were washed three times and lysed with 0.3% Triton X-100 (in PBS). Survived *S. aureus* in macrophages were plating serial dilutions on tryptic soy agar (in triplicate) of cell lysate. The colonies were counted after incubation at 37°C for 12–16 hr through CFUs.

## Lysosomal intracellular activity assay

The lysosomal activity of macrophages was measured using a Lysosomal Intracellular Activity Assay Kit (BioVision, K448-50) according to the manufacturer's protocol. Macrophages ($3×10^5$/well) were seeded in the six-well plates and incubate for 12 hr in DMEM medium supplemented with 10% FBS at 37°C with 5% $CO_2$, and the medium was replaced with fresh medium containing *S. aureus* (MOI 10). Bafilomycin A1 was added in the negative control group. Cells were incubated for 1, 2, 4, 8 hr at 37°C with 5% $CO_2$, then washed three times with PBS. Cells were treated with medium containing gentamicin (200 μg/ml) and Self-Quenched Substrate for 1 hr at 37°C with 5% $CO_2$. Cells were harvested and washed twice with 1 ml ice-cold Assay Buffer, and then re-suspended in 1 ml of PBS. Cells were analyzed by flow cytometer (488 nm excitation laser).

## Transmission electron microscopy

The macrophages were treated with *S. aureus* at an MOI of 10 for 4 hr. Cells were washed three times with PBS and collected to measure the autophagy level by TEM. Macrophages were fixed in 3% glutaraldehyde plus 2% paraformaldehyde for 24 hr and then in 1% osmic acid for 1 hr. Subsequently, the procedures were conducted using the standard protocol. After dehydrating in a graded series of ethanol, cells were embedded in epoxyresin. The thin sections were observed under a JEM-1230 microscope at 80 kV.

## Measurement of cAMP

The level of cAMP was measured by the enzyme-linked immunosorbent assay (ELISA) using an ELISA kit (Mlbio, ml061899) according to the manufacturer's protocol. Cells were lysed in RIPA buffer containing protease inhibitors and phosphatase inhibitors. Cell lysis was centrifugated at the speed of 2000 rpm for 20 min. Supernatant was added into the sensitive spectrophotometric 96 plate to combine with HRP-labeled detection antibodies. 3,3',5,5'-Tetramethylbenzidine was used as the chromogenic substrate for HRP. The optical density was measured at 450 nm with a microplate analyzer.

## Computational processing and bioinformatics of RNA-seq

For the RNA-seq alignment, purified sequence data was aligned to reference goat genome downloaded from Ensembl website (https://asia.ensembl.org/index.html) by HISAT2. Read count matrices were obtained using Feature Counts. DEGs were accessed using R package DESeq2, with a false discovery rate less than 0.05 and |log2foldchange|>2 or 3. PCA was also performed by DESeq2. Both GO and KEGG functional enrichment analysis or visualization were finished by R package ClusterProfiler or ClueGO, a plug-in of Cytoscape. Additionally, autophagy-related gene sets were obtained from Molecular Signatures Database (MSigDB) (http://www.broadinstitute.org).

## Statistical analysis

All data were analyzed by the Student's t test. Data are expressed as mean ± standard deviations (SD). Statistical significance was marked as '*' when $p<0.05$, '**' when $p<0.01$, and '***' when $p<0.001$. The statistical difference was considered significant at a level of $p<0.05$. Plotting and correlation statistical tests were performed on R v3.6.3. Dot plots were generated using the R ggplot2 function. Heatmaps were drew using R package ComplexHeatmap.

## Acknowledgements

This work was supported by the National Transgenic Creature Breeding Grand Project of China (2016Z × 08008-003).

## Additional information

### Funding

| Funder | Grant reference number | Author |
|---|---|---|
| National Transgenic Key Project of the Ministry of Agriculture of China | 2016ZX08008-003 | Hongbing Han |

The funders had no role in study design, data collection and interpretation, or the decision to submit the work for publication.

### Author contributions

Mengyao Wang, Data curation, Formal analysis, Investigation, Writing - original draft, Writing – review and editing; Yu Qi, Data curation, Formal analysis, Investigation, Writing – review and editing; Yutao Cao, Formal analysis, Visualization; Xiaosheng Zhang, Yongsheng Wang, Qingyou Liu, Jinlong Zhang, Investigation; Guangbin Zhou, Yue Ai, Shao Wei, Linli Wang, Writing – review and editing; Guoshi Liu, Zhengxing Lian, Conceptualization; Hongbing Han, Conceptualization, Data curation, Funding acquisition, Project administration, Supervision, Writing - original draft, Writing – review and editing

### Author ORCIDs

Mengyao Wang (ID) http://orcid.org/0000-0002-1186-1792
Guangbin Zhou (ID) http://orcid.org/0000-0002-4493-8311
Hongbing Han (ID) http://orcid.org/0000-0002-6671-719X

### Ethics

All animal experiments and treatments were approved and supervised by the Animal Welfare Committee of China Agricultural University (Approval ID: AW31501202-1-1).

### Decision letter and Author response

Decision letter https://doi.org/10.7554/eLife.78044.sa1
Author response https://doi.org/10.7554/eLife.78044.sa2

## Additional files

### Supplementary files

• Supplementary file 1. Quantification and statistical analysis.

• Supplementary file 2. The data for generation of clone goat and part primers used in this study. (A) Table displaying the generation of clone goats by nuclear transfer. (B) Primers of crRNA-oligo. (C) Table displaying the sequences of primers in T7 endonuclease 1 (T7E1) assay. (D) All primers used for PCR. (E) Table displaying the primers for real-time reverse transcription PCR (qRT-PCR).

• Transparent reporting form

### Data availability

All data generated or analyzed during this study are included in the manuscript and supporting files or Dryad Dataverse. GO and KEGG terms analysis files have been provided for Figures 1H, 4B and 4C.

The following dataset was generated:

| Author(s) | Year | Dataset title | Dataset URL | Database and Identifier |
|---|---|---|---|---|
| Wang M, Han H | 2022 | GO and KEGG terms analysis files between TLR2-4 and WT cells | https://dx.doi.org/10.5061/dryad.mw6m905zk | Dryad Digital Repository, 10.5061/dryad.mw6m905zk |

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
