## [Editor Report]

In this manuscript, the authors generate a goat expressing a domain fusion receptor TLR2-4 that allows macrophages from the genetically modified goat to eliminate *Staphylococcus aureus*. This study is of interest to animal geneticists studying molecular breeding for infection resistance and individuals interested in bacterial infections. These studies also serve as a good animal model for disease resistance breeding.

---

## [Decision Letter]

**Decision letter after peer review:**

Thank you for submitting your article "Domain fusion TLR2-4 enhances the autophagy-dependent clearance of *Staphylococcus aureus* in the genetic engineering goat generated by CRISPR/Cas9" for consideration by *eLife*. Your article has been reviewed by 3 peer reviewers, and the evaluation has been overseen by a Reviewing Editor and Tadatsugu Taniguchi as the Senior Editor. The following individual involved in the review of your submission has agreed to reveal their identity: Kui Li (Reviewer #3).

Essential revisions:

– In this manuscript, all the experiments to prove that TLR2-4 enhanced *S. aureus*-induced autophagy and explore which signaling pathways played a role in this process were based on in vitro (fibroblasts and macrophages). Did the authors conduct in vivo experiments to verify the corresponding mechanism?

– L. 106-109. In Figure 1A, there was an unmentioned TLR2 signal domain flanking the MYC-Tag sequence. Please indicate the purpose of TLR2 signal domain construct.

– L. 109-113. The authors described that sgRNA2 was the most favorable target by comprehensive analysis of the genome editing efficiency. Why not use the most efficient sgRNA3 (28.45% indels in Figure S1A) for the further TLR2-4 knock-in, and instead use the sgRNA2 (16.69%) with relatively moderate efficiency?

– L. 119-120. In Figure S1C, there were weak and miscellaneous bands. How did the authors judge which cell colonies were TLR2-4 integrated?

– L. 131-133. According to Figure 1F-G, compared to WT fibroblast cells with siTLR2, the mRNA level of IL-8 in siTLR2-treated TLR2-4 fibroblasts was dramatically increased with heat-killed *S. aureus* treatment. However, the mRNA level of IL-8 in TLR2-4 fibroblasts with siTLR2 was significantly decreased with P2CSK4 and P3CSK4 treatment. Please check this and discuss.

– L. 170. In Figure 2F, please check whether the housekeeping gene used is GAPDH (legend) or β-actin (diagram)? Keep diagram and legend consistent.

– In this paper, why was heat-killed S. aureus used in some experiments (L. 125-133, L. 135-138, L. 144-148, etc.) and live *S. aureus* was used in others (L. 174-176, L. 180-183, L. 187-189, etc.)? Was there any difference between them? Please describe and discuss.

– L. 180-183. In Figure S3D, there was a huge difference in infection rates of *S. aureus* at MOI=10 between WT1 and WT2. Please explain and discuss.

– L. 199-201 and L. 205-207. In the absence of S. aureus treatment, why did WT macrophages show significantly higher LC3-I expression than TLR2-4 macrophages (Figure 3F and 3G)? But in L. 201-204, when *S. aureus* was not treated, why did TLR2-4 macrophages show significantly higher LC3-I expression than WT macrophages (Figure S3G)? Please describe and discuss.

– L. 220-222. The authors described that the lysosomal activity of WT macrophages was similar to that of the TLR2-4 transgenic macrophages treated with Bafilomycin A1. So, in Figure S3H, were "TLR2-4 + S. aureus" and "TLR2-4 + Bafilomycin A1 + *S. aureus*" marked incorrectly?

– These abbreviations should be spelled out at first use: "TLR" (L. 29), "WT" (L. 130), "HK-SA" (Figure 1I), "WB" (L. 398). It would be good to check the whole manuscript to make sure abbreviations are used correctly.

– The loading control, GAPDH, seems to change due to the treatment (as shown in Figures 5A, B and D). This raises the question of whether the quantitative data for the target protein is reliable. Do the authors have another loading control, or can they also use total protein?

– For some Figures (i.e. Figures 5A, B and D) the horizontally indicated group of WT contains some of the TLR2-3 experimental group. The full name and abbreviation in the legend are used at the same time and it would be recommended to be more uniform (Figure 5). DNA size of the ladder should also be marked in Figure 2F, etc.

– On Page 16, lines 329-331: the authors mentioned that: conversion of LC3-I to LC3-II was increased in TLR2-4 macrophages with *S. aureus* treatment by activator Forskolin, while was decreased by inhibitor H-89. However, the LC3-I and LC3-II patterns didn't show an obvious difference between Forskolin and H-89-treated groups (Figure 6G), thus, the authors need to repeat this experiment and/or interpret the data accurately.

– The two RNA-seq results were not compared, resulting in a lack of overall understanding of the total number of differentially expressed genes (DEGs). In lines 136-138, 557 DEGs were detected between TLR2-4 and WT fibroblast cells. Similarly, in lines 235-237, 1310 DEGs were detected between TLR2-4 and WT macrophages. So what's the number of common and unique genes for these two experiments? What are the possible reasons for the difference between the two results?

– In line 291-292, the author mentioned that previous studies demonstrated the activation of *JNK* and ERK signaling promotes the expression of ATG5 and ATG12. This conclusion seems not to be supported by this research, as the author came to a conclusion that inhibition of *JNK* and ERK signaling increased the expression of the ATG5 and ATG12 (lines 292-296). For this divergence, the author has not made further elaboration and discussion, nor has it been reflected in the subsequent Figure 7. The authors should make sure their data is reproducible and comment on these divergent findings.

– Please emphasize the application and significance of this study in the Discussion section. As an opportunist pathogen, *S. aureus* causes a range of serious diseases.

– Line 161, what is STED5? Please clarify when it appears for the first time.

– Line 207-211, please describe the results of Figure 3G in detail to better connect the subsequent suggestion.

– The authors only mentioned MyD88 in the abstract, introduction and Figure 7, but did not mention or examine it in the main text. Please clarify.

*Reviewer #2 (Recommendations for the authors):*

The conclusions of this paper are mostly well supported by data, but some aspects of image acquisition and data analysis need to be clarified and extended.

1) The description of the results is inaccurate and inconsistent with the Figures. For example, on page 7, lines 131-133, the authors described that the mRNA level of IL-6 and IL-8 in TLR2-4 fibroblast cells with siTLR2 was dramatically increased with P2CSK4, P3CSK4 and heat-killed S. aureus treatment, receptively (Figure 1F, G). However, as shown in Figures 1F and G, the mRNA level of IL-8 in TLR2-4 fibroblast cells with siTLR2 was dramatically decreased with P2CSK4, P3CSK4 treatment, while only increased with heat-killed *S. aureus* treatment. In addition, the mRNA level of IL1B was also inconsistent across treatment groups, which the authors did not explain in detail. The authors should factually describe the results and explain them reasonably.

2) The data on the activity of TLR2-4 presented in Figure 1I is not sufficiently convincing. For example, What's the meaning of "HK-SA" in Figure 1I? On page 7, lines 149-150, the authors described that the level of NF-κB subunit p65 translocation was almost suppressed completely in TLR2-4 fibroblasts independent of *S. aureus* treatment. The authors should provide data to support this conclusion. Numbering-Terms, symbols, and abbreviations are numbered and are not in accordance with a standard format. For example, WTsiTLR2+P2C, TLR2-4 siTLR2+P2C; WT siTLR2+P3C, TLR2-4 siTLR2+P3C, what are the meaning of P2C and P3C?

3) The method described is unclear, siRNA only knocks down endogenous TLR2, does it also affect TLR2-4 (Page 6, lines 126-127)?

4) The number of experimental animals is relatively small. The authors describe that only one clone goat was obtained (Figure 2A). Dose all of the macrophages were from this clone goat? Adding more animals experiments to assess individual differences would be needed to increase confidence in the presented reliable results. It would be more valuable to provide in vivo experiments such as the elimination of *S. aureus* in TLR2-4 overexpressing goats.

*Reviewer #3 (Recommendations for the authors):*

In general, the key claims of the manuscript are well supported by the data and the approaches are properly used. But some aspects of data analysis need to be extended.

(1) The manuscript focuses on the role of autophagy to eliminate *S. aureus* intracellular. In Figure 3E, the authors mentioned that the autophagosomes in TLR2-4 macrophages were more than in WT macrophages. However, to better make results reliable, I suggest quantifying the proportion of autophagosomes on a statistical graph.

(2) Western blot experiments in Figure 1I, Figure 5A and D, Figure 6D, it would be better if the changes of blots were shown using the ratio to housekeeping protein.

(3) The keywords should be ordered according to their relevance to the article as Toll-like receptor 2-4; autophagy; *Staphylococcus aureus*; macrophages; goat.

---

## [Author Response]

Essential revisions:– In this manuscript, all the experiments to prove that TLR2-4 enhanced *S. aureus*-induced autophagy and explore which signaling pathways played a role in this process were based on in vitro (fibroblasts and macrophages). Did the authors conduct in vivo experiments to verify the corresponding mechanism?

We would to thank the learned editor and reviewers for this valuable comment on our manuscript. We agree completely. Staphylococcus aureus (*S. aureus*) is the major contagious pathogen associated with goat mastitis. The severity of clinical *S. aureus* mastitis varies from mild, expressed only by changes in milk, to acute gangrenous mastitis causing necrosis of the affected udder quarter, severe systemic signs, and even death of goat. It leads to reduced milk production and huge economic losses. Due to the widespread use of antibiotic, *S. aureus* has developed antibiotic resistance mechanisms for most kinds of antibiotics. Therefore, a novel strategy is needed to control *S. aureus* infection.

The TLR2-4 cloned goat we obtained was male, which needs to be bred with wild-type females for related studies on mastitis. However, the generation interval of goats was long, it will take about three years to obtain sexually mature female cloned goats expressing TLR2-4. So, it takes a long time to conduct vivo experiments to study the mechanism on resistance to *S. aureus* mastitis. Further, our main aim is to explore the domain fusion TLR2-4 receptor could enhance autophagy-mediated *S. aureus* clearance in macrophages. The autophagy events still need to be verified at the cellular level, even though vivo experiments were carried out. At present, conclusions we obtained through cell experiments are well supported by data. We are planning to carry out the next in vivo experiments, and continue to investigate the resistance of TLR2-4 transgenic goats to *S. aureus*-induced mastitis in the future.

– L. 106-109. In Figure 1A, there was an unmentioned TLR2 signal domain flanking the MYC-Tag sequence. Please indicate the purpose of TLR2 signal domain construct.

The recombinant receptor contained the exon sequence of TLR2 extracellular domain including its own signal peptide to direct target protein secretion and across membranes. We have mentioned TLR2 signal domain in the manuscript on page 6, line 111.

– L. 109-113. The authors described that sgRNA2 was the most favorable target by comprehensive analysis of the genome editing efficiency. Why not use the most efficient sgRNA3 (28.45% indels in Figure S1A) for the further TLR2-4 knock-in, and instead use the sgRNA2 (16.69%) with relatively moderate efficiency?

Thank you for carefully reading. According to the editing efficiency that evaluated by the T7E1 assay, we selected targets sgRNA2, sgRNA3 and sgRNA7 for subsequent clonal cells screening. During the screening process, the clonal cells targeted by sgRNA2 had stronger proliferative ability than cells with other targets. The ultimate goal is to screen out clonal cells that have inserted the *TLR2-4* gene. Therefore, we have selected the sgRNA2-targeted clonal cells with better growth status for further identification.

– L. 119-120. In Figure S1C, there were weak and miscellaneous bands. How did the authors judge which cell colonies were TLR2-4 integrated?

Thank you for the suggestion. We lysed half of the clonal cells in the 96-well plate as a template for PCR, and the other half were continued to culture. The TLR2-4-specific band (1420 bp) was obtained from PCR assay. We have marked the DNA size of the ladder in Figure 1—figure supplement 1C. According to the size of DNA ladder, we selected the clonal cells with the expected band. After expanding and culturing these clones, the genomic DNA was extracted as PCR templates, and the DNA fragment of 1420 bp was further purified to sequence by Sanger sequencing (Figure 1—figure supplement 1C). The clones with correct sequencing result were judged to integrate TLR2-4.

– L. 131-133. According to Figure 1F-G, compared to WT fibroblast cells with siTLR2, the mRNA level of IL-8 in siTLR2-treated TLR2-4 fibroblasts was dramatically increased with heat-killed *S. aureus* treatment. However, the mRNA level of IL-8 in TLR2-4 fibroblasts with siTLR2 was significantly decreased with P2CSK4 and P3CSK4 treatment. Please check this and discuss.

We appreciate the Reviewer’s suggestion. It is a dynamic process that cells produce cytokines. It has been reported overexpression of TLR4 only promoted the production of proinflammatory cytokines at the early stage of LPS stimulation. After LPS treatment for more than 4 h, the TLR4-mediated TRIF-dependent pathway was activated, and the cytokines secreted by TLR4-overexpressing macrophages were reduced compared with WT macrophages (Wei et al., 2019). As previously reported, the mRNA level of IL-6 in TLR2-4 fibroblast cells with siTLR2 was dramatically increased, while IL-8 and IL-1β were significantly decreased after stimulated by TLR2 ligands for 10 h. Therefore, we hypothesized that the expression of IL-8 and IL-1β mediated by TLR2-4 may be inhibited through TRIF-induced internalization pathway. In addition, it has been reported pretreatment of microglia with P3C induces the secretion of anti-inflammatory cytokine and attenuates inflammatory response (Kochan et al., 2012). These results demonstrated TLR2-4 recognizes ligands by the extracellular domain of TLR2 and initiates downstream immune response by the intracellular domain of TLR4.

P3CSK4 (P3C) is a synthetic triacylated lipopeptide and a TLR2/TLR1 ligand. P2CSK4 (P2C) is a synthetic diacylated lipopeptide and is recognized by the TLR2/TLR6 heterodimer. They mimic the acylated amino terminus of bacterial lipopeptides and are potent activator of the pro-inflammatory transcription factor NF-κB and AP-1. However, *S. aureus* is the whole cell pathogen with PAMPs on surface that include lipoteichoic acid (LTA) and peptidoglycan (PGN) in addition to lipopeptides. LTA and PGN have been reported to induce the release of IL-8 through MyD88-dependent signaling pathway (Hattar et al., 2017; Zenhom et al., 2012). Therefore, the mRNA levels of downstream cytokines were different between Figure 1F and Figure 1G. We have revised the Discussion section. Please see the text on pages 18-19, lines 369-389.

– L. 170. In Figure 2F, please check whether the housekeeping gene used is GAPDH (legend) or β-actin (diagram)? Keep diagram and legend consistent.

Thank you for highlighting this mistake. The housekeeping gene is β-actin. We have made corrections in figure legend.

– In this paper, why was heat-killed S. aureus used in some experiments (L. 125-133, L. 135-138, L. 144-148, etc.) and live *S. aureus* was used in others (L. 174-176, L. 180-183, L. 187-189, etc.)? Was there any difference between them? Please describe and discuss.

Thank you for the suggestion. In our study, heat-killed *S. aureus* was used in the experiments in fibroblasts, and live *S. aureus* was used in experiments in macrophages. Since fibroblasts are non-immune cells, heat-killed *S. aur*eus was used to simulate the ligands that activate membrane receptors on non-phagocytotic cells to investigate the signal conduct pathways in immune responses. While macrophages are phagocytotic immune cells, the challenge of live *S. aureus* was used to explore the signaling pathways that are activated during phagocytosis and elimination of *S. aureus* by immune cells.

– L. 180-183. In Figure S3D, there was a huge difference in infection rates of *S. aureus* at MOI=10 between WT1 and WT2. Please explain and discuss.

We agree completely with the reviewer’s comment. We speculated that the difference in infection rate between WT1 and WT2 was caused by the difference in TLR2 expression level on the surface of WT macrophages. In order to make sure the reproducibility of the data, we have repeated this experiment. Please see the revised Figure 2—figure supplement 1D.

– L. 199-201 and L. 205-207. In the absence of S. aureus treatment, why did WT macrophages show significantly higher LC3-I expression than TLR2-4 macrophages (Figure 3F and 3G)? But in L. 201-204, when *S. aureus* was not treated, why did TLR2-4 macrophages show significantly higher LC3-I expression than WT macrophages (Figure S3G)? Please describe and discuss.

Thank you for your suggestion. It was found that the LC3I expression in untreated macrophages and fibroblasts were different, which suggested that TLR2-4 might affect the expression of LC3I in different cells. In addition, it has been shown that cells in the two experiments often resulted in the different levels of LC3I expression, even in cells of the same strain (Fei et al., 2020). In the Figure 3F and 3G, macrophages were infected by live *S. aureus*, while in Figure 2—figure supplement 1H, fibroblasts were stimulated by HK-SA, which could result in different expression levels of LC3I in the two experiments. We have made corrections in the revised manuscript. Please see the text on page 11, lines 209-211.

– L. 220-222. The authors described that the lysosomal activity of WT macrophages was similar to that of the TLR2-4 transgenic macrophages treated with Bafilomycin A1. So, in Figure S3H, were "TLR2-4 + *S. aureus*" and "TLR2-4 + Bafilomycin A1 + *S. aureus*" marked incorrectly?

Thank you for highlighting this mistake. The legends in Figure S3H were mislabeled. We have made corrections in revised Figure 2—figure supplement 1I.

– These abbreviations should be spelled out at first use: "TLR" (L. 29), "WT" (L. 130), "HK-SA" (Figure 1I), "WB" (L. 398). It would be good to check the whole manuscript to make sure abbreviations are used correctly.

Thank you for bringing this to our attention. We have carefully revised the abbreviations in this manuscript.

– The loading control, GAPDH, seems to change due to the treatment (as shown in Figures 5A, B and D). This raises the question of whether the quantitative data for the target protein is reliable. Do the authors have another loading control, or can they also use total protein?

Thank you for the valuable suggestion. GAPDH, initially identified as a glycolytic enzyme and considered as a housekeeping gene, is widely used as internal controls for gene expression normalization for western blotting in macrophages (Fang et al., 2021; Laha et al., 2019). Therefore, we selected *GAPDH* as housekeeping gene. However, the bicinchoninic colorimetric (BCA) assay was sometimes insufficient for accurate total protein quantification resulting in changes in the reference protein. In order to clearly display the proportional change of the target genes, we have added the ratios of target genes to housekeeping genes in Figures. In addition, we found that *GAPDH* in the first panel of Figure 5A changed after treatment. *Tubulin* has replaced *GAPDH* as housekeeping gene now. Please see the revised Figure 5.

– For some Figures (i.e. Figures 5A, B and D) the horizontally indicated group of WT contains some of the TLR2-3 experimental group. The full name and abbreviation in the legend are used at the same time and it would be recommended to be more uniform (Figure 5). DNA size of the ladder should also be marked in Figure 2F, etc.

Thank you for careful reading. Since the names of inhibitors are relatively long, we use abbreviations to make the legends clearer in Figure 5. Among them, endocytosis inhibitor Dynasore, *TAK1* inhibitor Takinib, and TBK1 inhibitor Amlexanox, are generally not represented by abbreviations in the articles (Mowers et al., 2013; Panipinto et al., 2021; Zhong et al., 2019). Therefore, we have kept the full names for the convenience of readers to understand better. We have marked DNA size of the ladder in Figure 2F, Figure 1—figure supplement 1A and Figure 1—figure supplement 1C. Please see the revised Figures.

– On Page 16, lines 329-331: the authors mentioned that: conversion of LC3-I to LC3-II was increased in TLR2-4 macrophages with *S. aureus* treatment by activator Forskolin, while was decreased by inhibitor H-89. However, the LC3-I and LC3-II patterns didn't show an obvious difference between Forskolin and H-89-treated groups (Figure 6G), thus, the authors need to repeat this experiment and/or interpret the data accurately.

Thank you for the suggestion. LC3I to LC3II conversion on western blot analysis is now widely used to monitor autophagy. We have marked the ratios of LC3II to LC3I in Figure 6G. As shown in revised Figure 6G, cAMP upregulation by activator Forskolin resulted in enhancement of LC3-II to LC3-I ratio in *S. aureus*-infected TLR2-4 macrophages. Further, pretreated with the protein kinase A inhibitor H-89 significantly reduced the ratio of LC3-II to LC3-I in *S. aureus*-infected TLR2-4 macrophages but not in the *S. aureus*-infected wide type macrophages. These data indicated that TLR2-4 enhanced *S. aureus*-induced autophagy in macrophages via cAMP-PKA signaling.

– The two RNA-seq results were not compared, resulting in a lack of overall understanding of the total number of differentially expressed genes (DEGs). In lines 136-138, 557 DEGs were detected between TLR2-4 and WT fibroblast cells. Similarly, in lines 235-237, 1310 DEGs were detected between TLR2-4 and WT macrophages. So what's the number of common and unique genes for these two experiments? What are the possible reasons for the difference between the two results?

Thank you for this valuable suggestion. In the two RNA sequencing experiments, cells were processed differently. In Figure 1—figure supplement 2A, both *S. aureus*-infected TLR2-4 and WT fibroblasts were pretreated with siRNA targeting endogenous TLR2 to detect whether TLR2-4 we transfected could recognize *S. aureus*. In the second RNA-seq experiment, TLR2-4 and WT macrophages were directly treated with *S. aureus*. Therefore, the number of DEGs were different in the two experiments.

– In line 291-292, the author mentioned that previous studies demonstrated the activation of JNK and ERK signaling promotes the expression of ATG5 and ATG12. This conclusion seems not to be supported by this research, as the author came to a conclusion that inhibition of JNK and ERK signaling increased the expression of the ATG5 and ATG12 (lines 292-296). For this divergence, the author has not made further elaboration and discussion, nor has it been reflected in the subsequent Figure 7. The authors should make sure their data is reproducible and comment on these divergent findings.

We agree with the reviewer’s comment. We have repeated this experiment to ensure the reproducibility of the data, and the results are shown in the following figure. The expression of ATG5 and ATG12 was substantially increased by *JNK* or ERK inhibitor after *S. aureus* treatment in both TLR2-4 macrophages and the WT macrophages. We have added a comment to this result in the Discussion section and presented in Figure 7. Please see the revised text on page 21, lines 423-429, and the revised Figure 7.

– Please emphasize the application and significance of this study in the Discussion section. As an opportunist pathogen, *S. aureus* causes a range of serious diseases.

The reviewer’s suggestion is important. We have emphasized the significance of this study in the Discussion section. Please see the revised text on page 18, lines: 366-368.

– Line 161, what is STED5? Please clarify when it appears for the first time.

Thank you for highlighting this mistake. “SETD5” was mistyped as “STED5”. We have made corrections and marked its full name in the text on page 7, lines: 115-116.

– Line 207-211, please describe the results of Figure 3G in detail to better connect the subsequent suggestion.

Thank you for the suggestion. We have clarified the results in Figure 3G. Please see the revised text on pages 11, lines: 211-223.

– The authors only mentioned MyD88 in the abstract, introduction and Figure 7, but did not mention or examine it in the main text. Please clarify.

Thank you for the valuable suggestion. Since many researches have demonstrated that the adaptor MyD88 is required for TLR2 and TLR4 signal transduction (Akira and Takeda, 2004; Fitzgerald et al., 2001), we examined the activation of downstream MAPKs and NF-κB. To determine the accuracy of our data, we have monitored the expression of MyD88 in TLR2-4 and WT macrophages, as shown in Figure 5—figure supplement 1A. The results were consistent with previous studies, which found that the expression of MyD88 in TLR2-4 macrophages was significantly higher than that in WT macrophages.

Reviewer #2 (Recommendations for the authors):The conclusions of this paper are mostly well supported by data, but some aspects of image acquisition and data analysis need to be clarified and extended.1) The description of the results is inaccurate and inconsistent with the Figures. For example, on page 7, lines 131-133, the authors described that the mRNA level of IL-6 and IL-8 in TLR2-4 fibroblast cells with siTLR2 was dramatically increased with P2CSK4, P3CSK4 and heat-killed S. aureus treatment, receptively (Figure 1F, G). However, as shown in Figures 1F and G, the mRNA level of IL-8 in TLR2-4 fibroblast cells with siTLR2 was dramatically decreased with P2CSK4, P3CSK4 treatment, while only increased with heat-killed *S. aureus* treatment. In addition, the mRNA level of IL1B was also inconsistent across treatment groups, which the authors did not explain in detail. The authors should factually describe the results and explain them reasonably.

We appreciate the Reviewer’s comments. We have carefully determined the comments, and all suggestions made of reviewer were complied with during the revision.

Thank you for the valuable suggestion. We have redescribed the results of Figure 1F, G and explained them in the Discussion section. Please see the revised text on pages 18-19, lines 369-389.

2) The data on the activity of TLR2-4 presented in Figure 1I is not sufficiently convincing. For example, What's the meaning of "HK-SA" in Figure 1I? On page 7, lines 149-150, the authors described that the level of NF-κB subunit p65 translocation was almost suppressed completely in TLR2-4 fibroblasts independent of *S. aureus* treatment. The authors should provide data to support this conclusion. Numbering-Terms, symbols, and abbreviations are numbered and are not in accordance with a standard format. For example, WTsiTLR2+P2C, TLR2-4 siTLR2+P2C; WT siTLR2+P3C, TLR2-4 siTLR2+P3C, what are the meaning of P2C and P3C?

Thank you for the suggestion. We have carefully revised the abbreviations in our manuscript. Please see the revised text on page 7, lines 133-135. HK-SA: heat-killed *S. aureus*; P2C: P2CSK4; P3C: P3CSK4. The data of NF-κB subunit p65 translocation was shown in Figure 1I. Thank you for bring this to our attention. We have linked the figure in the revised text on page 8, line 156.

3) The method described is unclear, siRNA only knocks down endogenous TLR2, does it also affect TLR2-4 (Page 6, lines 126-127)?

Thank you for careful reading. The siRNA did not affect TLR2-4 because it targets the intracellular domain of endogenous TLR2. TLR2-4 is assembled the extracellular domain of TLR2 with the transmembrane and intracellular domains of TLR4. We have supplemented this information in the Materials and methods section. Please see the revised text on page 31, lines 568-569.

4) The number of experimental animals is relatively small. The authors describe that only one clone goat was obtained (Figure 2A). Dose all of the macrophages were from this clone goat? Adding more animals experiments to assess individual differences would be needed to increase confidence in the presented reliable results. It would be more valuable to provide in vivo experiments such as the elimination of *S. aureus* in TLR2-4 overexpressing goats.

We agree completely with the Reviewer’s comment. All of the macrophages were from this TLR2-4 clone goat. *S. aureus* is the major contagious pathogen associated with goat mastitis. The TLR2-4 cloned goat we obtained was male, which needs to be bred with wild-type females for related studies on mastitis. However, the generation interval of goats was long, it will take about three years to obtain sexually mature female cloned goats expressing TLR2-4. So, it takes a long time to conduct vivo experiments to study the mechanism on resistance to *S. aureus* mastitis. Further, our main aim is to explore the domain fusion TLR2-4 receptor could enhance autophagy-mediated *S. aureus* clearance in macrophages. The autophagy events still need to be verified at the cellular level, even though vivo experiments were carried out. At present, conclusions we obtained through cell experiments are well supported by data. We are planning to carry out the next in vivo experiments, and continue to investigate the elimination of *S. aureus* in TLR2-4 overexpressing goats and the resistance of TLR2-4 overexpressing goats to *S. aureus*-induced mastitis in the future.

Reviewer #3 (Recommendations for the authors):In general, the key claims of the manuscript are well supported by the data and the approaches are properly used. But some aspects of data analysis need to be extended.(1) The manuscript focuses on the role of autophagy to eliminate *S. aureus* intracellular. In Figure 3E, the authors mentioned that the autophagosomes in TLR2-4 macrophages were more than in WT macrophages. However, to better make results reliable, I suggest quantifying the proportion of autophagosomes on a statistical graph.

We appreciate the Reviewer’s valuable suggestions. We have carefully made corrections and examined the manuscript.

The reviewer’s point is important and we have added a statistical graph in Figure 2—figure supplement 1G. Please see the revised Figure 2—figure supplement 1G.

(2) Western blot experiments in Figure 1I, Figure 5A and D, Figure 6D, it would be better if the changes of blots were shown using the ratio to housekeeping protein.

Thank you for the valuable suggestion. We have optimized the results of western blots to make it clearer. The expression levels of target genes have been normalized with the ratio of housekeeping genes. Please see the revised Figure 1I, Figure 5A, Figure 5D, and Figure 6D.

(3) The keywords should be ordered according to their relevance to the article as Toll-like receptor 2-4; autophagy; *Staphylococcus aureus*; macrophages; goat.

Thank you for careful reading. We have revised the Keywords. Please see the revised keywords on page 2, line 41.

References

Akira S, and Takeda K. 2004. Toll-like receptor signalling. *Nat Rev Immunol* 4: 499-511. DOI:10.1038/nri1391

Fang S, Wan X, Zou X, Sun S, Hao X, Liang C, Zhang Z, Zhang F, Sun B, Li H, and Yu B. 2021. Arsenic trioxide induces macrophage autophagy and atheroprotection by regulating ROS-dependent TFEB nuclear translocation and AKT/mTOR pathway. *Cell Death Dis* 12: 88. DOI:10.1038/s41419-020-03357-1

Fei Q, Ma H, Zou J, Wang W, Zhu L, Deng H, Meng M, Tan S, Zhang H, Xiao X, Wang N, and Wang K. 2020. Metformin protects against ischaemic myocardial injury by alleviating autophagy-ROS-NLRP3-mediated inflammatory response in macrophages. *J Mol Cell Cardiol* 145: 1-13. DOI:10.1016/j.yjmcc.2020.05.016

Fitzgerald KA, Palsson-McDermott EM, Bowie AG, Jefferies CA, Mansell AS, Brady G, Brint E, Dunne A, Gray P, Harte MT, McMurray D, Smith DE, Sims JE, Bird TA, and O'Neill LA. 2001. Mal (MyD88-adapter-like) is required for Toll-like receptor-4 signal transduction. *Nature* 413: 78-83. DOI:10.1038/35092578

Hattar K, Reinert CP, Sibelius U, Gökyildirim MY, Subtil FSB, Wilhelm J, Eul B, Dahlem G, Grimminger F, Seeger W, and Grandel U. 2017. Lipoteichoic acids from *Staphylococcus aureus* stimulate proliferation of human non-small-cell lung cancer cells in vitro. *Cancer Immunol Immunother* 66: 799-809. DOI:10.1007/s00262-017-1980-4

Kochan T, Singla A, Tosi J, and Kumar A. 2012. Toll-like receptor 2 ligand pretreatment attenuates retinal microglial inflammatory response but enhances phagocytic activity toward *Staphylococcus aureus*. *Infect Immun* 80: 2076-2088. DOI:10.1128/iai.00149-12

Laha D, Deb M, and Das H. 2019. KLF2 (kruppel-like factor 2 [lung]) regulates osteoclastogenesis by modulating autophagy. *Autophagy* 15: 2063-2075. DOI:10.1080/15548627.2019.1596491

Mowers J, Uhm M, Reilly SM, Simon J, Leto D, Chiang SH, Chang L, and Saltiel AR. 2013. Inflammation produces catecholamine resistance in obesity via activation of PDE3B by the protein kinases IKKε and TBK1. *eLife* 2: e01119. DOI:10.7554/*eLife*.01119

Panipinto PM, Singh AK, Shaikh FS, Siegel RJ, Chourasia M, and Ahmed S. 2021. Takinib Inhibits Inflammation in Human Rheumatoid Arthritis Synovial Fibroblasts by Targeting the Janus Kinase-Signal Transducer and Activator of Transcription 3 (JAK/STAT3) Pathway. *Int J Mol Sci* 22:10.3390/ijms222212580

Wei S, Yang D, Yang J, Zhang X, Zhang J, Fu J, Zhou G, Liu H, Lian Z, and Han H. 2019. Overexpression of Toll-like receptor 4 enhances LPS-induced inflammatory response and inhibits *Salmonella* Typhimurium growth in ovine macrophages. *European journal of cell biology* 98: 36-50. DOI:10.1016/j.ejcb.2018.11.004

Zenhom M, Hyder A, de Vrese M, Heller KJ, Roeder T, and Schrezenmeir J. 2012. Peptidoglycan recognition protein 3 (PglyRP3) has an anti-inflammatory role in intestinal epithelial cells. *Immunobiology* 217: 412-419. DOI:10.1016/j.imbio.2011.10.013

Zhong B, Shi D, Wu F, Wang S, Hu H, Cheng C, Qing X, Huang X, Luo X, Zhang Z, and Shao Z. 2019. Dynasore suppresses cell proliferation, migration, and invasion and enhances the antitumor capacity of cisplatin via STAT3 pathway in osteosarcoma. *Cell Death Dis* 10: 687. DOI:10.1038/s41419-019-1917-2